# Allomorphy as a mechanism of post-translational control of enzyme activity

Henry P. Wood[1,4], F. Aaron Cruz-Navarrete [1,4✉], Nicola J. Baxter[1,2,4], Clare R. Trevitt[1], Angus J. Robertson[1,3], Samuel R. Dix [1], Andrea M. Hounslow[1], Matthew J. Cliff [2] & Jonathan P. Waltho [1,2✉]

Enzyme regulation is vital for metabolic adaptability in living systems. Fine control of enzyme activity is often delivered through post-translational mechanisms, such as allostery or allokairy. β-phosphoglucomutase (βPGM) from *Lactococcus lactis* is a phosphoryl transfer enzyme required for complete catabolism of trehalose and maltose, through the isomerisation of β-glucose 1-phosphate to glucose 6-phosphate via β-glucose 1,6-bisphosphate. Surprisingly for a gatekeeper of glycolysis, no fine control mechanism of βPGM has yet been reported. Herein, we describe allomorphy, a post-translational control mechanism of enzyme activity. In βPGM, isomerisation of the K145-P146 peptide bond results in the population of two conformers that have different activities owing to repositioning of the K145 sidechain. In vivo phosphorylating agents, such as fructose 1,6-bisphosphate, generate phosphorylated forms of both conformers, leading to a lag phase in activity until the more active phosphorylated conformer dominates. In contrast, the reaction intermediate β-glucose 1,6-bisphosphate, whose concentration depends on the β-glucose 1-phosphate concentration, couples the conformational switch and the phosphorylation step, resulting in the rapid generation of the more active phosphorylated conformer. In enabling different behaviours for different allomorphic activators, allomorphy allows an organism to maximise its responsiveness to environmental changes while minimising the diversion of valuable metabolites.

[1] Krebs Institute for Biomolecular Research, Department of Molecular Biology and Biotechnology, The University of Sheffield, Sheffield S10 2TN, UK. [2] Manchester Institute of Biotechnology and School of Chemistry, The University of Manchester, Manchester M1 7DN, UK. [3] Present address: Laboratory of Chemical Physics, National Institute of Diabetes and Digestive and Kidney Diseases, National Institutes of Health, Bethesda, Maryland 20892, USA. [4] These authors contributed equally: Henry P. Wood, F. Aaron Cruz-Navarrete, Nicola J. Baxter. ✉email: facruznavarrete1@sheffield.ac.uk; j.waltho@sheffield.ac.uk

Enzyme regulation is vital in maintaining the balance of catabolism and anabolism in living systems[1–3]. Enzyme activity is subject to precise control, sometimes involving manifold layers of regulation, and failure often results in metabolic disorders and disease[4,5]. Regulatory mechanisms are divided into two broad categories: those relating to the control of enzyme concentration (coarse control) and those that modulate enzyme activity (fine control). In coarse control, concentration is determined by transcriptional modulation of gene expression and the balance between the rates of translation and degradation, with additional contributions from maturation, cellular compartmentalisation and local co-clustering[6–9]. Coarse control occurs on relatively long timescales (hours to days). In fine control, a diverse group of regulatory mechanisms act to modulate enzyme activity over much shorter timescales (<second to minutes). This group includes the binding of regulatory molecules and reversible covalent modification[10,11], and often involves allosteric modulation, where an effector, acting somewhere other than the active site, stabilises forms of the enzyme with a reduced or enhanced activity[2,12–14]. Alternatively, allokairy is a fine control mechanism, where the activity of a monomeric enzyme is modulated by the near-equivalence of the conformational exchange rate and the catalytic rate in a substrate concentration-dependent manner[15,16].

Precise enzyme regulation allows organisms to be responsive to environmental changes and to exploit multiple energy sources. *Lactococcus lactis* (*L. lactis*) is a Gram-positive bacterium that has worldwide usage in the manufacture of fermented dairy products and in the commercial production of lactic acid[17]. It can grow on a variety of carbohydrate media including trehalose and maltose[18–20]. Trehalose is transported into *L. lactis* by the phosphoenolpyruvate-dependent phosphotransferase system, yielding trehalose 6-phosphate (T6P), which is phosphorolysed by $P_i$-dependent trehalose 6-phosphate phosphorylase to β-glucose 1-phosphate (βG1P) and glucose 6-phosphate (G6P)[21] (Supplementary Fig. 1). In contrast, maltose enters cells by the ATP-dependent permease system and is phosphorolysed by the action of $P_i$-dependent maltose phosphorylase to βG1P and glucose[22]. Glucose is subsequently phosphorylated to G6P by glucokinase and enters glycolysis via fructose 1,6-bisphosphate (F16BP). For complete catabolism of both trehalose and maltose, the isomerisation of βG1P to G6P is catalysed by β-phosphoglucomutase (βPGM, EC 5.4.2.6, 25 kDa). βPGM-deficient *L. lactis* is unable to grow or has impaired growth, when the sole carbon source is trehalose or maltose, respectively[23]. With maltose, βG1P accumulates intracellularly and is

excreted into the growth medium. Correspondingly, both $P_i$-dependent trehalose 6-phosphate phosphorylase and $P_i$-dependent maltose phosphorylase (Supplementary Fig. 1) operate in the reverse sense to their physiological roles in wild-type *L. lactis*, resulting in βG1P being combined with G6P to form T6P or polymerised to form amylose (α(1–4)-linked glucose units). In trehalose and maltose metabolism, therefore, βPGM acts as the gatekeeper to and from glycolysis, and is expected to be subject to tight regulation. In terms of coarse control, transcription of the βPGM gene (*pgmB*), which is located in the *tre* operon, is subject to negative transcriptional control by glucose and lactose[19]. When *L. lactis* switches from metabolising glucose to metabolising maltose (or by implication, trehalose), there is a significant rise in the specific activity of βPGM over a period of several hours. However, no fine control mechanism has yet been identified at basal levels of βPGM, which would allow the cell to compete more successfully during a transition between carbohydrate sources.

βPGM is a monomeric magnesium-dependent phosphoryl transfer enzyme of the haloacid dehalogenase (HAD) superfamily[24–31]. The active site is located in the cleft between the α/β core domain (M1–D15, S88–K216) and the α-helical cap domain (T16–V87), with closure of the cleft through domain reorientation occurring during catalysis. Two phosphate group binding sites are present, a proximal site adjacent to the carboxylate nucleophile and the catalytic $Mg^{2+}$ ion, and a distal site located ~8 Å away in the closed enzyme[29]. During steady-state catalysis, βG1P binds to phosphorylated βPGM (βPGM$^P$, phosphorylated on D8) and forms β-glucose 1,6-bisphosphate (βG16BP). Release to solution and subsequent rebinding of βG16BP in the alternate orientation[32] leads to the formation of G6P and the regeneration of βPGM$^P$ (Fig. 1). In vitro, a phosphorylating (priming) agent is required to initiate the catalytic cycle since the half-life of βPGM$^P$ is ~30 s[28]. In vivo, potential candidates for this agent include F16BP, βG1P, G6P, α-glucose 1,6-bisphosphate (αG16BP) and acetyl phosphate (AcP), as well as the reaction intermediate, βG16BP. However, only βG16BP allows βPGM to reach its maximum catalytic rate, and a significant lag phase is observed in the reaction with αG16BP as the phosphorylating agent[28], until the βG16BP concentration greatly exceeds its resting concentration in the cell. In the current kinetic model for βPGM catalysis, αG16BP is also required to act as a very strong inhibitor of βPGM. αG16BP is a close structural analogue of βG16BP[25], but very similar kinetic behaviour is observed when AcP is used as the phosphorylating agent[31], suggesting that other factors are con-

**Fig. 1 βPGM catalytic cycle.** βPGM reaction scheme for the enzymatic conversion of βG1P to G6P via a βG16BP intermediate. The phosphoryl transfer reaction between phospho-enzyme (βPGM$^P$, phosphorylated at residue D8) and βG1P is illustrated with the transferring phosphate (blue) in the proximal site and the 1-phosphate (red) of βG1P in the distal site. The phosphoryl transfer reaction between βPGM and βG16BP is shown with the transferring phosphate (red) in the proximal site and the 6-phosphate (blue) of βG16BP in the distal site.

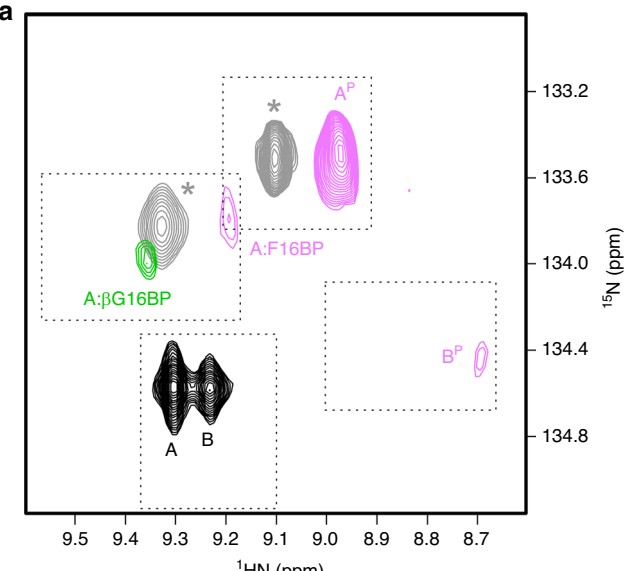

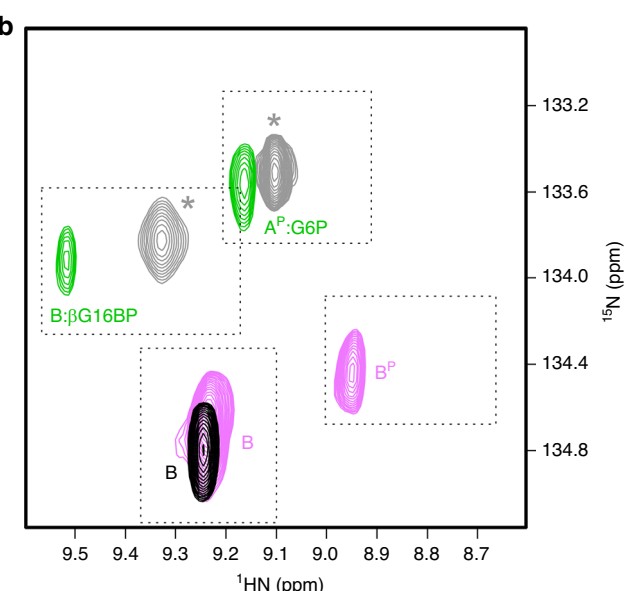

**Fig. 2 Effect of different phosphorylating agents on βPGM. a, b** Overlays of a section of $^1H^{15}N$-TROSY spectra highlighting the behaviour of residue A113. **a** βPGM$_{WT}$ (black) populates conformer A and conformer B in slow exchange. βPGM$_{WT}$ supplemented with F16BP (pink) populates phosphorylated conformer A ($A^P$) as the dominant species, phosphorylated conformer B ($B^P$) and a βPGM$_{WT}$:F16BP species (A:F16BP). βPGM$_{WT}$ supplemented with βG16BP (green) populates an A:βG16BP complex. **b** βPGM$_{P146A}$ (black) populates one conformer (conformer B). βPGM$_{P146A}$ supplemented with F16BP (pink) populates conformer B and $B^P$. βPGM$_{P146A}$ supplemented with βG16BP (green) populates a $A^P$:G6P complex and a B:βG16BP complex. Peaks indicated by grey asterisks correspond to the βPGM$_{WT}$:BeF$_3$ complex (grey; $\delta_N$ = 133.5 ppm; BMRB 17851[35]), which is an analogue of $A^P$, and the Mg$^{2+}$-saturated βPGM$_{D10N}$:βG16BP complex (grey; $\delta_N$ = 133.8 ppm; BMRB 27174[31]), which is a mimic of the A:βG16BP complex, and are shown for comparison.

tributing to post-translational control of βPGM. Here we show, through combined use of NMR spectroscopy, X-ray crystallography, site-directed mutagenesis and kinetic assays, that a regulatory mechanism is operating in βPGM, which we have termed allomorphy to illustrate its relationship to and distinction from allostery and allokairy. In the substrate-free enzyme, the

isomerisation of proline 146 results in the near-equal population of two conformers that have different activities. Alternative phosphorylating agents such as F16BP and AcP generate phosphorylated forms of both conformers, resulting in a lag phase in βPGM activity until the more active phosphorylated conformer dominates. In contrast, the βG16BP reaction intermediate is able to couple the conformational switch and the phosphorylation step, resulting in the rapid generation of the more active phosphorylated species. This allows the βG16BP concentration to effectively act as a surrogate of the βG1P concentration and modulate the activity of βPGM according to the carbohydrate source available to *L. lactis*.

## Results

**βPGM$_{WT}$ exchanges slowly between two stable conformations.** The observation of a lag phase when using either αG16BP or AcP[28,31] as the phosphorylating agent implies that the target of phosphorylation, the substrate-free enzyme, has a role in post-translational control. Hence, the solution properties of substrate-free wild-type βPGM (βPGM$_{WT}$) were investigated using NMR spectroscopy. In the previous backbone resonance assignment of βPGM$_{WT}$ (BMRB 7235[26]) performed in standard NMR buffer (50 mM K$^+$ HEPES (pH 7.2), 5 mM MgCl$_2$, 2 mM NaN$_3$, 10% (v/v) $^2H_2O$ and 1 mM trimethylsilyl propanoic acid (TSP)) containing 10 mM NH$_4$F, two features were apparent during the analysis: (1) peaks of 30 active site residues were missing from the spectra owing to line-broadening resulting from conformational exchange on the millisecond timescale and (2) a large number of unassigned $^1H_N$, $^{15}N$, $^{13}C\alpha$, $^{13}C\beta$ and $^{13}C'$ resonances were present with a low intensity. To test whether HEPES or NH$_4$F were contributing to the millisecond conformational exchange, spectra were recorded in tris buffer (50 mM tris (pH 7.2), 5 mM MgCl$_2$, 2 mM NaN$_3$, 10% (v/v) $^2H_2O$ and 1 mM TSP) and mixtures of HEPES and tris buffers in order to transfer the assignment between conditions[33]. It was noticed that the inclusion of 5 mM tris in the standard NMR buffer increased the intensity of the unassigned resonances significantly and therefore all observable resonances in the spectra were re-assigned using standard triple resonance TROSY-based methodology[34]. Excluding the ten proline residues and the N-terminal methionine, the backbone resonances of 193 out of a total of 210 residues (92%) were assigned. Seventeen residues located in the vicinity of the active site remained unassigned (L9, D10, G11, R38, L44, K45, G46, S48, R49, E50, D51, S52, L53, K117, N118, D170 and S171). Notably, 102 of the assigned residues displayed pairs of resonances in the $^1H^{15}N$-TROSY spectrum (Fig. 2a, Supplementary Fig. 2a), consistent with the population of two βPGM$_{WT}$ conformers (70% conformer A, BMRB 28095 and 30% conformer B, BMRB 28096). A further five residues (K145, A147, D149, I150 and Q176) have assignments in conformer A, but are missing assignments in conformer B, owing to some differential millisecond conformational exchange occurring in the two species. The βPGM$_{WT}$ conformers are present in the spectra as a result of slow conformational exchange rather than as chemically distinct species, as the addition of 3 mM BeCl$_2$ and 10 mM NH$_4$F to the βPGM$_{WT}$ sample induced the population of a single βPGM$_{WT}$:BeF$_3$ complex (an analogue of phosphorylated conformer A; BMRB 17851[35]) (Supplementary Fig. 3a). The exchange between conformer A and conformer B is on the multi-second timescale, with $k_{ex} \leq 1.0 \, s^{-1}$ from ZZ-exchange measurements. Differences in chemical shift between the two conformers (Supplementary Fig. 4a) indicate that the regions of βPGM$_{WT}$ involved in the multi-second conformational exchange process are located primarily in the core domain and comprise the D137–A147 loop, the β-strands (K109–A113 and D133–A136) at the outer edge of the

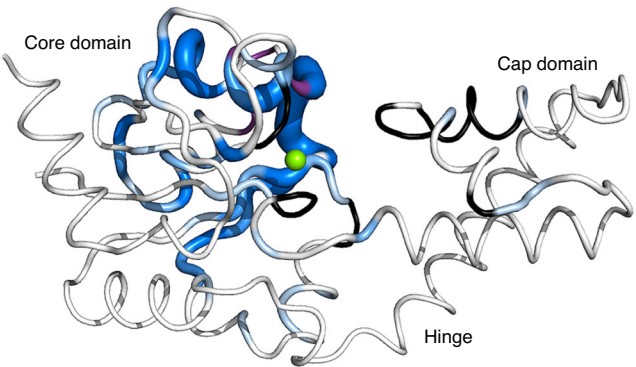

**Fig. 3 Exchange behaviour in βPGM$_{WT}$.** Crystal structure of βPGM$_{WT}$ (PDB 2WHE[29]) showing residues of βPGM$_{WT}$ undergoing conformational exchange on different timescales. Residues that populate two conformations in slow exchange are coloured in shades of blue according to chemical shift differences between conformer A and conformer B, with the intensity of colour and thickness of the backbone corresponding to larger values. Residues in conformer A and conformer B with missing backbone amide peaks in the $^1$H$^{15}$N-TROSY spectrum of βPGM$_{WT}$ are coloured black, whereas missing backbone amide peaks in conformer B only are coloured purple. The amide $^1$H$^{15}$N coherences are likely broadened beyond detection owing to intermediate exchange on the millisecond timescale. The catalytic Mg$^{2+}$ ion is indicated as a green sphere.

β-sheet and the I152–S163 and Q172–A183 α- and 3$_{10}$-helical regions (Fig. 3). Predicted random coil index order parameters (RCI-S$^2$)[36] show a decrease in value for conformer B in two regions (G32–R38 in the cap domain and D133–K145 in the core domain) (Supplementary Fig. 5a), which indicates increased conformational flexibility compared with conformer A.

**Influence of physiological factors on the conformational exchange.** An investigation of factors that could potentially affect the population distribution of conformer A and conformer B was performed using $^1$H$^{15}$N-TROSY spectra of βPGM$_{WT}$ recorded under different conditions of temperature, pH, hydrostatic pressure, MgCl$_2$ (0–100 mM), NaCl (0–200 mM), K$^+$ HEPES buffer (0–200 mM) and βPGM$_{WT}$ concentration (0.1–1.2 mM). All of these perturbations had little or no effect, apart from the addition of either MgCl$_2$ (100 mM) or NaCl (200 mM) to standard NMR buffer, which shifted the population of βPGM$_{WT}$ primarily to conformer A (Supplementary Fig. 6a–d). Buffer exchange into deionised water resulted in conformer B being the dominant population. However, both conformer A and conformer B remained populated when Mg$^{2+}$ was removed from the NMR buffer solution, showing that the multi-second conformational exchange is not simply a result of incomplete saturation of the catalytic Mg$^{2+}$-binding site. These observations indicate that chloride anions perturb the population distribution.

The inorganic ionic composition of *L. lactis* cytoplasm (~2 mM Mg$^{2+}$, ~50 mM Na$^+$, ~400 mM K$^+$, ~50 mM Cl$^-$)[37] overlaps with the concentration ranges tested, where the population distribution between conformer A and conformer B remained unaffected. Therefore, it is expected that both conformer A and conformer B are populated in cytoplasm. However, the intracellular milieu is a complex mix of metabolites that could influence this equilibrium. This environment was mimicked through the use of bovine skimmed milk, a medium in which *L. lactis* thrives within the dairy industry. It is anticipated that the organic components in milk will also be present in cytoplasm. Moreover, the inorganic ionic composition (~5 mM Mg$^{2+}$, ~24 mM Na$^+$, ~38 mM K$^+$, ~28 mM Cl$^-$)[38] is similar to cytoplasm (except for K$^+$, which has no effect on the equilibrium between

conformer A and conformer B), so any effects will be owing to the influence of metabolites. βPGM$_{WT}$ was diluted fivefold into fresh skimmed milk, which had been filtered to remove species with a molecular weight larger than 10 kDa. The $^1$H$^{15}$N-TROSY spectrum revealed that both conformer A and conformer B were populated with a similar ratio (60% conformer A and 40% conformer B) to βPGM$_{WT}$ recorded in standard NMR buffer (Supplementary Figs. 2c, 7a). However, minor chemical shift changes in the active site loops and the sharpening of some peaks that were line-broadened under standard conditions indicated that one of the milk components was binding in the vicinity of the active site. The two dominant organic components of the filtered milk were lactose and citrate (Supplementary Fig. 2d). Titration of lactose into βPGM$_{WT}$ had no effect on the $^1$H$^{15}$N-TROSY spectrum, whereas titration of citrate led to equivalent chemical shift changes and sharpening of line-broadened peaks to those observed in milk. Similar effects were observed in both conformer A and conformer B. Hence, βPGM$_{WT}$ was crystallised in the presence of citrate and the structure was determined to 2.1 Å resolution (PDB 6YDM; Supplementary Fig. 8a–c, Supplementary Table 1). Two chains are present in the crystallographic asymmetric unit, one of which has citrate and acetate bound, whilst the other has tris and acetate bound. Citrate is coordinated in the active site by residues T16, H20, V47–R49 and A115–K117 and mimics substrate binding to some extent. Both monomers share a similar fold and overlay closely with a previously reported substrate-free βPGM$_{WT}$ structure (PDB 2WHE[29]; non-H atom RMSDs of 0.56 Å and 0.95 Å). Although only one of the two conformers observed in solution is represented in the crystal, the NMR experiments show that both conformer A and conformer B remain well-populated under physiological conditions.

**The conformational exchange involves cis–trans proline isomerisation.** Exchange phenomena on multi-second timescales in proteins are often a consequence of *cis–trans* isomerisation of Xaa-Pro peptide bonds[39,40]. The largest differences in chemical shift between conformer A and conformer B are observed for residues in a loop (D137–A147) containing two proline residues (P138 and P146) (Supplementary Fig. 4a). From the crystal structures of the substrate-free form of the enzyme (PDB 6YDL (Supplementary Table 1) determined to 1.5 Å resolution, which compares closely with PDB 2WHE[29] (non-H atom RMSD = 0.53 Å) and PDB 1ZOL[25] (non-H atom RMSD = 0.65 Å)), nine *trans* Xaa-Pro peptide bonds are present in βPGM$_{WT}$, whereas the K145–P146 peptide bond adopts a *cis* conformation. Proline residues with *cis* peptide bonds have $^{13}$Cβ nuclei that resonate 2.0–2.5 ppm downfield from those with *trans* peptide bonds[41] and therefore the isomerisation state of the Xaa-Pro peptide bonds for βPGM$_{WT}$ in solution was investigated. All but one of the assigned proline residues in conformer A and conformer B possess $^{13}$Cβ chemical shifts in the range 30.4–31.9 ppm consistent with the population of *trans* Xaa-Pro peptide bonds (Supplementary Fig. 5b). In contrast, the $^{13}$Cβ chemical shift for P146 (34.7 ppm) corroborates the presence of a *cis* K145–P146 peptide bond in solution for conformer A. However for conformer B, the absence of proline $^{13}$Cβ resonances for P146 and P148, owing to millisecond conformational exchange in the K145–I150 region, precluded an identification of the isomerisation state for these proline residues using NMR methods.

To explore whether proline isomerisation at the K145–P146 peptide bond is the source of the multi-second conformational exchange in βPGM$_{WT}$, the βPGM variant P146A (βPGM$_{P146A}$) was prepared and the solution properties of the substrate-free form were investigated. A $^1$H$^{15}$N-TROSY spectrum shows that only a single species is present (Fig. 2b, Supplementary Fig. 2b)

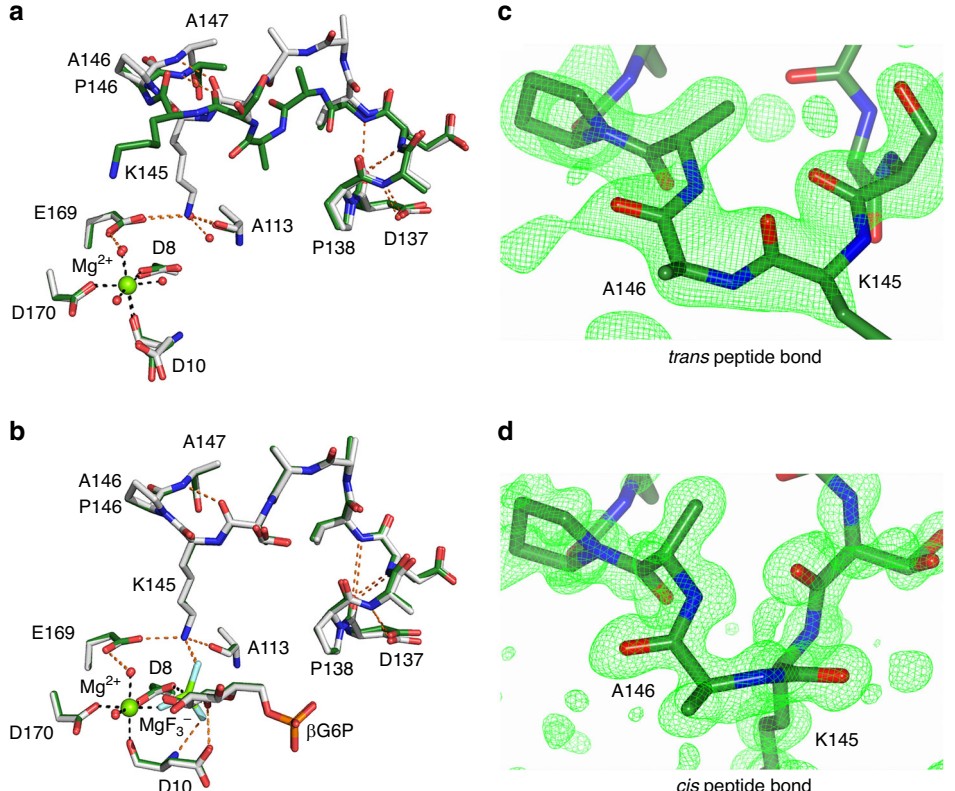

**Fig. 4 Conformational plasticity of the active site of βPGM. a, b** Active sites of βPGM$_{WT}$ (as conformer A) and βPGM$_{P146A}$ superposed on the core domain. **a** Selected residues are shown as sticks for the crystal structures of βPGM$_{WT}$ (grey carbon atoms; PDB 6YDL) and βPGM$_{P146A}$ (dark green carbon atoms; PDB 6YDK). In βPGM$_{WT}$, a *cis* K145–P146 peptide bond allows coordination of the K145 sidechain by E169 and A113, whereas in βPGM$_{P146A}$ a *trans* K146-A146 peptide bond changes significantly the backbone conformation of the D137–A147 loop, which precludes active site engagement of the K145 sidechain. The catalytic Mg$^{2+}$ ion is drawn as a green sphere, black dashes indicate metal ion coordination and orange dashes show probable hydrogen bonds. **b** Selected residues, the MgF$_3^-$ moiety and G6P are shown as sticks for the crystal structures of the βPGM$_{WT}$:MgF$_3$:G6P TSA complex (grey carbon atoms; PDB 2WF5[29]) and the βPGM$_{P146A}$:MgF$_3$:G6P TSA complex (dark green carbon atoms; PDB 6YDJ). βPGM$_{WT}$ maintains the *cis* K145–P146 peptide bond, whereas βPGM$_{P146A}$ changes the isomerisation state of the K145–A146 peptide bond from a *trans* conformation in the substrate-free enzyme to a *cis* conformation in the transition state. **c, d** Omit maps generated by refinement in the absence of residues S144–P148 in βPGM$_{P146A}$. **c** The S144–P148 segment, containing a *trans* K145–A146 peptide bond, with positive difference density (Fo–Fc; green mesh contoured at +2.5σ) in substrate-free βPGM$_{P146A}$. **d** The S144–P148 segment, containing a *cis* K145–A146 peptide bond, with positive difference density (Fo–Fc; green mesh contoured at +2.5σ) in the βPGM$_{P146A}$:MgF$_3$:G6P TSA complex.

and 194 out of a total of 211 residues (92%) were assigned using standard TROSY-based methodology (BMRB 27920[42]). The same seventeen residues as βPGM$_{WT}$ remain unassigned owing to millisecond conformational exchange. The chemical shifts of βPGM$_{P146A}$ were compared with those of conformer A and conformer B of βPGM$_{WT}$ (Supplementary Fig. 4b, c). Although the largest perturbations relate to the mutation site (together with an associated propagation of effects through the P148–V158 and S171–G182 α-helices), additional significant and widespread chemical shift differences are present between conformer A and βPGM$_{P146A}$, especially in the D137–A147 loop. In contrast, much smaller chemical shift changes are observed between conformer B and βPGM$_{P146A}$, indicating that the solution conformations for these species are closely similar. However, although the K145–A146 peptide bond in βPGM$_{P146A}$ is likely to adopt a *trans* conformation as the dominant population, the isomerisation state remains ambiguous using NMR methods. Therefore, βPGM$_{P146A}$ was crystallised and the structure was determined to 2.0 Å resolution (PDB 6YDK; Supplementary Table 1). The cap and the core domains of the crystal structure superimpose closely with those of βPGM$_{WT}$ (non-H atom RMSD = 0.33 Å, PDB 2WHE[29]; non-H atom RMSD = 0.48 Å, PDB 6YDL; non-H atom RMSD = 0.51 Å, PDB 1ZOL[25]). The D137–A147 loop exhibits

elevated temperature factors, consistent with the lower predicted RCI-S$^2$ values derived from NMR chemical shifts (Supplementary Fig. 5c). The electron density is best fit by the *trans* conformation of the K145–A146 peptide bond (ω dihedral angle = −177°) (Fig. 4a, c). In comparison with βPGM$_{WT}$, the D137–A147 loop adopts a different conformation, although both a 3$_{10}$-helix (D137–V141) and a β-turn hydrogen bond (A147$_{HN}$–S144$_{CO}$) are retained. These perturbations in structure are consistent with the chemical shift changes observed between conformer A and βPGM$_{P146A}$ and support the βPGM$_{P146A}$ structure being a close model of conformer B.

The most pronounced consequence of the change in isomerisation state of the K145–A146 peptide bond is the failure of the K145 sidechain in βPGM$_{P146A}$ to engage in the active site (Fig. 4a). Instead, this sidechain is positioned in the open cleft between the cap and core domains, and is exposed to solvent. In βPGM$_{WT}$, the ε-amino group of K145 is coordinated by the carbonyl oxygen atom of A113, the carboxylate sidechain of E169, and a water molecule that is replaced in the transition state analogue (TSA) complex (PDB 2WF5[29]) by a fluoride ion that mimics an oxygen atom of the transferring phosphoryl group. An electrostatic relationship also exists between the ε-amino group and the carboxylate group of D8. In βPGM$_{P146A}$,

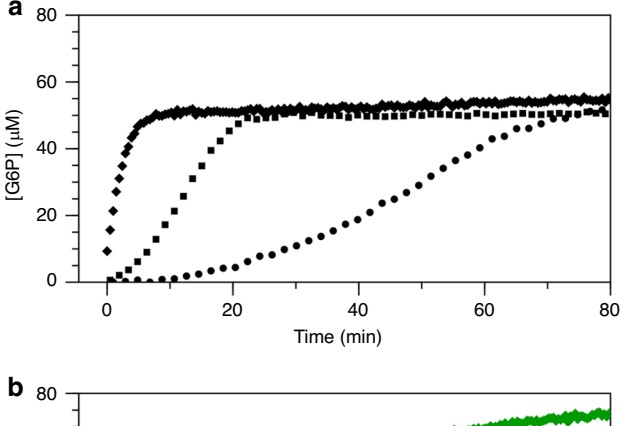

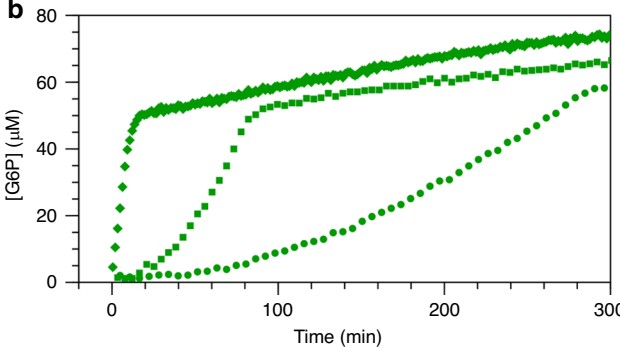

**Fig. 5 Kinetic profiles of βPGM activity. a, b** Reaction kinetics for the conversion of βG1P to G6P catalysed by βPGM_{WT} and βPGM_{P146A}. The rate of G6P production was measured indirectly using a glucose 6-phosphate dehydrogenase coupled assay, in which G6P is oxidised and concomitant $NAD^+$ reduction is monitored by the increase in absorbance at 340 nm. Reaction catalysed by either **a** βPGM_{WT} or **b** βPGM_{P146A} in standard kinetic buffer using either F16BP (circles), AcP (squares) or βG16BP (diamonds) as a phosphorylating agent. For clarity, between 100 and 8% of the data points are included in the kinetic profiles. Following βG1P substrate depletion, the kinetic profiles show an apparent increase in G6P concentration, which results from: (1) the concentration of the reaction ingredients through evaporation from the assay plate wells and (2) for the reactions recorded using βG16BP, the enzyme-dependent conversion of remaining βG16BP to G6P via βPGM^P, occurring at a rate proportional to the amount of enzyme.

the position of the missing ε-amino group of K145 and its βPGM_{WT} hydrogen bonding are satisfied by a water molecule. The predicted RCI-$S^2$ order parameters for βPGM_{P146A} and βPGM_{WT} (Supplementary Fig. 5a, c) share similar profiles apart from the D137–A147 loop region, where the RCI-$S^2$ values for βPGM_{P146A} indicate increased dynamic properties that broadly mirror those of conformer B in βPGM_{WT}. Together, these data reveal that βPGM_{P146A} reflects the properties of conformer B, and link the chemical shift and RCI-$S^2$ differences between conformers to the isomerisation state of the K145-X146 peptide bond. Thus, the multi-second exchange between conformer A and conformer B in solution involves *cis–trans* proline isomerisation of the K145–P146 peptide bond.

**βPGM_{WT} lag phase depends on the phosphorylating agent.** To ensure that the extent of the lag phase observed previously with AcP is not a method dependent observation[31], the effect of different phosphorylating agents on the mutase activity of βPGM_{WT} was investigated by monitoring the conversion of 50 μM βG1P to G6P with either F16BP (1 mM), AcP (8 mM) or βG16BP (10 μM) present as phosphorylating agents, using the standard glucose 6-phosphate dehydrogenase coupled assay[28]. Despite the crucial involvement of βG16BP as the reaction intermediate in the

catalytic cycle, its concentration in the cell can vary markedly and is dependent upon the concentration of βG1P. Therefore, F16BP is the most likely phosphorylating agent of βPGM_{WT} in vivo when *L. lactis* is growing on glucose-rich media (~50 mM F16BP[17] versus $K_m$ ~100 μM[25]). AcP is also a potential activator in vivo, as although it is present at lower concentrations (1–3 mM in *Escherichia coli* (*E. coli*)[43,44] versus $K_m$ ~800 μM[25]), it is inherently a much faster phosphorylating agent. In the coupled assay experiments with βPGM_{WT} (Fig. 5a), when either F16BP or AcP was used as the phosphorylating agent, their progression curves display significant lag phases. The lag is considerably more pronounced in the F16BP experiment, and consequently the maximum rate of βG1P to G6P conversion is not achieved before the substrate is exhausted. When βG16BP was used as the phosphorylating agent the kinetic profile shows a linear, fast initial rate. Consequently, initial rate measurements were made at several βG1P and βG16BP concentrations (10–700 μM and 0.4–100 μM, respectively) and were globally fitted to an equation derived for a ping–pong mechanism with βG1P inhibition[28]. Accurate fits were obtained at βG16BP concentrations up to 10 μM, as above this concentration the model no longer describes the data. At elevated βG16BP concentrations, the back reaction from βG16BP to βG1P becomes significant, and the free βG16BP concentration is attenuated owing to a multimeric interaction between βG16BP and $Mg^{2+}$ ions[31]. Accordingly, the data above 10 μM βG16BP were omitted from the fitting. This analysis yielded values for $k_{cat}$ of 382 ± 12 $s^{-1}$, $K_m$ (βG1P) of 91 ± 4 μM, $K_m$ (βG16BP) of 8.5 ± 0.3 μM and $K_i$ (βG1P) of 1510 ± 100 μM (Supplementary Fig. 9a). These values are all higher than those previously reported[25,28] for βPGM_{WT} owing to the extension of the analysis to higher βG1P and $Mg^{2+}$ concentrations.

**βPGM catalysis utilises a *cis* K145-X146 peptide bond.** To assess whether βPGM is active as conformer B, the effect of different phosphorylating agents on the mutase activity of βPGM_{P146A} was investigated by monitoring the conversion of 50 μM βG1P to G6P with either F16BP (1 mM), AcP (8 mM) or βG16BP (10 μM) present as phosphorylating agents using the standard glucose 6-phosphate dehydrogenase coupled assay. As for βPGM_{WT}, the kinetic profiles for βPGM_{P146A} display significant lag phases with F16BP and AcP, whereas the progression curve with βG16BP shows a linear, fast initial rate (Fig. 5b). Consequently, initial rate measurements were made at several βG1P and βG16BP concentrations (5–500 μM and 2–100 μM, respectively) and were globally fit to the equation used for βPGM_{WT} above. For βPGM_{P146A}, the fitting yielded values for $k_{cat}$ of 19.2 ± 0.2 $s^{-1}$, $K_m$ (βG1P) of 157 ± 3 μM, $K_m$ (βG16BP) of 175 ± 3 μM and $K_i$ (βG1P) of 933 ± 32 μM (Supplementary Fig. 9b). In addition, the equilibration of βG1P and G6P for both βPGM_{WT} and βPGM_{P146A} was monitored by $^{31}P$ NMR spectroscopy using AcP as a phosphorylating agent (Supplementary Fig. 9c, d)[31]. The time courses show a similar overall profile together with the presence of the lag phase and subsequent fitting of the linear segments yielded a $k_{obs}$ of 70 ± 30 $s^{-1}$ for βPGM_{WT} and a $k_{obs}$ of 1.1 ± 0.2 $s^{-1}$ for βPGM_{P146A}. The variation between the kinetic parameters derived using the two methods is caused by inhibition resulting from different levels of phosphate-containing species present in the assays. However, the data clearly demonstrate that βPGM_{P146A} is active, with a ~20-fold reduction in $k_{cat}$, a ~21-fold increase in $K_m$ for βG16BP and a similar $K_m$ and $K_i$ for βG1P, when compared with βPGM_{WT}.

The mechanism of βPGM_{P146A} catalytic activity was explored by preparing a TSA complex containing $MgF_3^-$ and G6P[29,45] and the resulting βPGM_{P146A}:MgF_3:G6P TSA complex was investigated using NMR spectroscopy. The observed

[19]F chemical shifts are indistinguishable from those of the βPGM$_{WT}$:MgF$_3$:G6P TSA complex (Supplementary Fig. 9e, f) and a [1]H[15]N-TROSY spectrum peak comparison (BMRB 7234[26]) indicates an almost identical correspondence between frequencies. Such close agreement allowed a backbone resonance assignment (211 residues—100%) using 3D HNCACB and 3D HN(CA)CO spectra (BMRB 28097). Residues with the largest chemical shift differences between the βPGM$_{P146A}$:MgF$_3$:G6P and βPGM$_{WT}$:MgF$_3$:G6P TSA complexes are located within 4 Å of residue 146 (K145, A147 and A177) and within 5 Å of residue K145 (S48, V141 and A142) (Supplementary Fig. 4d). Taken together, these results confirm that βPGM$_{P146A}$ can assemble a stable and wild-type like βPGM$_{P146A}$:MgF$_3$:G6P TSA complex in solution. The βPGM$_{P146A}$:MgF$_3$:G6P TSA complex was crystallised and the structure was determined to 1.0 Å resolution (PDB 6YDJ; Supplementary Table 1). This complex superimposes very closely with the βPGM$_{WT}$:MgF$_3$:G6P TSA complex (non-H atom RMSD = 0.18 Å, PDB 2WF5[29]) and identifies both the positioning of the K145 sidechain in the active site and the *cis* K145–A146 peptide bond (ω dihedral angle = 14°; compared with ω dihedral angle = 12° for the K145–P146 peptide bond in the βPGM$_{WT}$:MgF$_3$:G6P TSA complex) (Fig. 4b, d). The 6-phosphate group of G6P is in the distal site and the trigonal MgF$_3^-$ moiety mimicking the transferring phosphoryl group is coordinated in the proximal site between D8 (atom Oδ1) and the 1-OH group of G6P. The donor–acceptor distance and the angle of alignment are 4.1 Å and 174°, respectively (compared with 4.3 Å and 176°, respectively for the βPGM$_{WT}$:MgF$_3$:G6P TSA complex). The catalytic Mg$^{2+}$ ion coordination also has comparable octahedral geometry to the βPGM$_{WT}$:MgF$_3$:G6P TSA complex and to substrate-free βPGM$_{P146A}$. Together, these data demonstrate that βPGM$_{P146A}$ is able to populate a *cis* K145–A146 peptide bond and achieve full domain closure with concomitant formation of transition state geometry. In addition, assuming that βG16BP binding is diffusion controlled, the increase in $K_m$ for βG16BP in βPGM$_{P146A}$ reflects the energetic cost of the *trans* to *cis* isomerisation of the K145–A146 peptide bond[46]. As in βPGM$_{WT}$, these results imply that conformer A of βPGM$_{P146A}$ represents the more active form.

**βPGM forms two different transient phospho-enzyme species.** The possible involvement of conformer B in the modulation of enzyme activity was investigated using real-time NMR methods by comparing the phosphorylation of βPGM under saturating conditions of either F16BP (50–100 mM), AcP (60–100 mM) or βG16BP (35 mM). Residue A113 is a well-resolved reporter of the relevant species—conformer A and conformer B, and their phosphorylated counterparts, A$^P$ and B$^P$. The carbonyl group of A113 is coordinated by the ε-amino group of K145 (in conformer A) or a water molecule (in conformer B) and its amide proton is hydrogen bonded to the carbonyl group of F7 (adjacent to the D8 nucleophile) (Fig. 4a, b).

On addition of F16BP to βPGM$_{P146A}$, the two dominant species observed are conformer B and B$^P$ (Fig. 2b). The presence of conformer B shows that the phosphorylation rate of βPGM$_{P146A}$ is very similar to the dephosphorylation rate of B$^P$ (through hydrolysis), and only an apparent rate constant can be measured. The apparent rate constant for dephosphorylation was determined to be 0.003 ± 0.00002 s$^{-1}$ from the rate of reduction of the free F16BP concentration in [1]H NMR experiments. The [1]H and [15]N chemical shifts of B$^P$, assigned using fast acquisition 3D HNCO and 3D HNCA NMR experiments, mirror those of conformer B, except for the active site residues F7–D8,

A113–A115 and hinge residues T16–E18, owing to their proximity to phosphorylated D8 (Supplementary Fig. 4e). Resonances from the D137–A147 loop show no significant differences between both forms, indicating that the K145–A146 peptide bond is in a *trans* conformation in B$^P$ (the conformer B to A$^P$ transition results in large chemical shift changes for the D137–A147 loop; Supplementary Fig. 4f). Conformer B and B$^P$ are also observed when AcP was used as the phosphorylating agent, and a minor population of A$^P$ is present, correlating with a small increase in the population of B$^P$ relative to conformer B (Supplementary Fig. 7b). Identification of A$^P$ is based on the assignment and structure of the βPGM$_{WT}$:BeF$_3$ complex (BMRB 17851; PDB 2WFA[35]), where the K145–P146 peptide bond is in a *cis* conformation and the K145 sidechain is engaged in the active site. Notably, when βG16BP was used as the phosphorylating agent, B$^P$ is not observed (Fig. 2b, Supplementary Fig. 7b). Instead, the A$^P$:G6P and B:βG16BP complexes are the primarily populated species. The A$^P$:G6P complex has similar chemical shifts to the βPGM$_{WT}$:BeF$_3$ complex, and the slow exchange between the B:βG16BP and the A$^P$:G6P complexes correlates with the measured $k_{cat}$ values for βPGM$_{P146A}$.

In βPGM$_{WT}$, A$^P$ is the dominant species observed on addition of F16BP (Fig. 2a, Supplementary Fig. 3c). Therefore, the phosphorylation rate of βPGM$_{WT}$ by F16BP under these conditions must be faster than the hydrolysis rate of A$^P$ ($k_{hydrolysis}$ = 0.06 ± 0.006 s$^{-1}$)[31]. A minor population of the A:F16BP complex is also present, indicating that the phosphorylation rate is slower than the chemical shift difference between the A$^P$ and A:F16BP peaks (140 Hz). Significantly, a minor population of B$^P$ is also observed. This species is populated transiently (~5 min) and disappears at longer timeframes, whereas A$^P$ and the A:F16BP complex populations remain dominant while the phosphorylating agent is at high concentration. Hence, the B$^P$ population is converting to the more stable A$^P$ species with a rate constant of ≥0.003 s$^{-1}$, which mirrors the *trans* to *cis* isomerisation rate constants of Xaa-Pro peptide bonds in model peptides[39]. Equivalent behaviour is observed when AcP was used as the phosphorylating agent (Supplementary Figs. 3b, 7a), except that an A:AcP complex is not detected. When βG16BP was used as a phosphorylating agent, B$^P$ does not accumulate at any point in the 3 h time course. The only detectable species is an A:βG16BP complex (Fig. 2a, Supplementary Figs. 3d, 7a), which is identified by the similarity of chemical shift distribution with the βPGM$_{D10N}$:βG16BP complex (BMRB 27174; PDB 5OK1[31]). The low intensity of the A113 peak, along with peaks of other active site residues (Fig. 2a, Supplementary Figs. 3d, 7a), results from millisecond conformational exchange between species within the catalytic cycle, which correlates with the measured $k_{cat}$ values for βPGM$_{WT}$. Overall therefore, the consequence of phosphorylation by the reaction intermediate βG16BP is markedly different to that of other phosphorylating agents, in that it generates no detectable B$^P$ or lag phase, even when the initial population of conformer B is high.

**Discussion**
Substrate-free βPGM$_{WT}$ exists in solution as two distinct conformers with near-equal populations, which differ in the isomerisation state of the K145–P146 peptide bond and interconvert at a rate between 0.003 s$^{-1}$ and 1.0 s$^{-1}$. Conformer A contains the *cis*-isomer of this peptide bond, as observed in the crystal structures of substrate-free βPGM$_{WT}$, whereas conformer B contains the *trans*-isomer, as mimicked by the βPGM$_{P146A}$ variant. In the crystal structure of βPGM$_{P146A}$, the *trans* K145–A146 peptide bond positions the sidechain of K145 away from the site of phosphoryl transfer, which is significantly different to its location

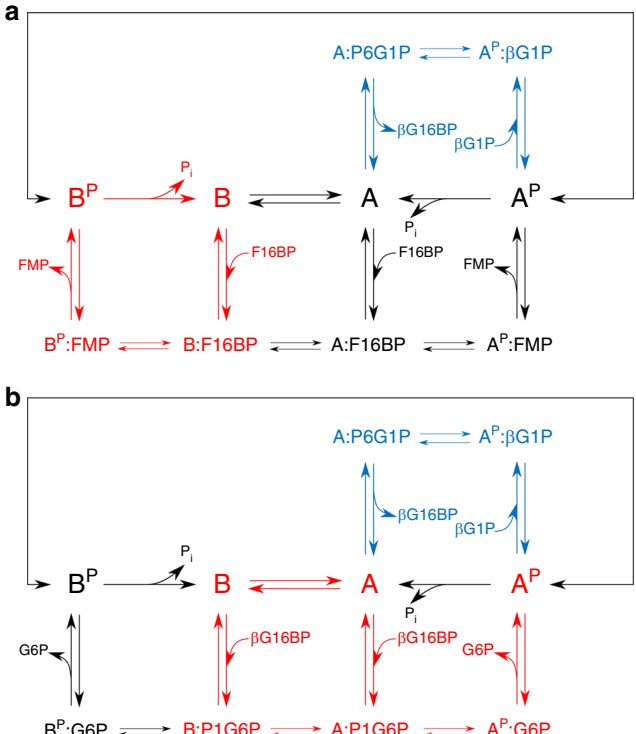

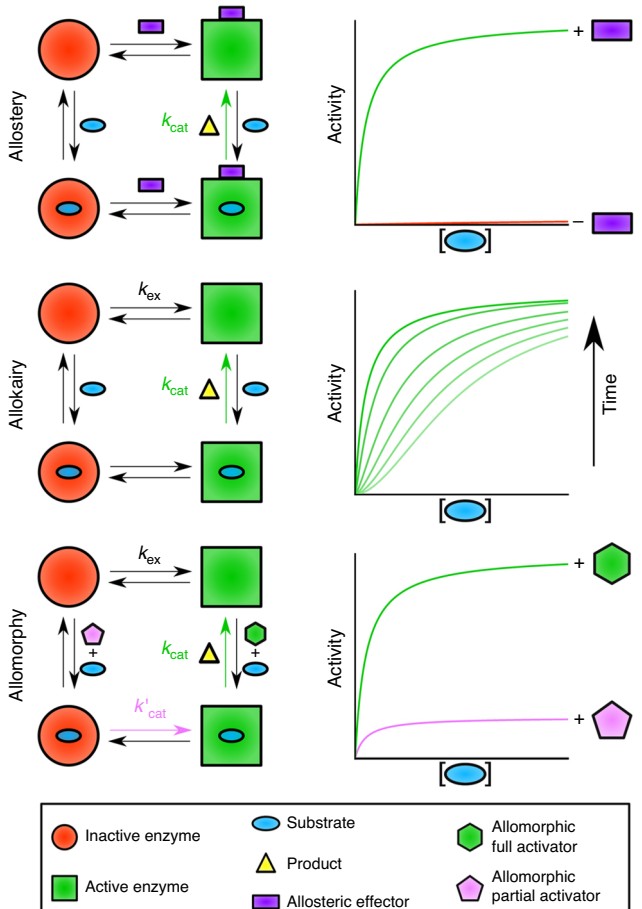

**Fig. 6 Kinetic model of βPGM activity. a, b** Reaction schemes for βPGM$_{WT}$ as conformer A or conformer B with different phosphorylating agents. The favoured pathways are shown (red text) for βPGM$_{WT}$ with **a** F16BP as a phosphorylating agent and **b** βG16BP as a phosphorylating agent. The βG16BP generating steps are highlighted in blue text. Fructose monophosphate (FMP) is either fructose 6-phosphate or fructose 1-phosphate. The complexes X:P1G6P (X = A or B) and A:P6G1P denote explicitly the orientation of βG16BP bound in the active site. The double-headed arrows connecting A$^P$ and B$^P$ indicate that these species interconvert with a multi-second exchange rate, similar to that described for the interconversion of conformer A and conformer B.

**Fig. 7 Mechanisms of regulation and activity profiles in monomeric enzymes.** In allostery, binding (or reaction) of an allosteric effector (purple rectangle) outside of the active site shifts the enzyme population from an inactive form (red circle and red profile) to an active form (green square and green profile), which stimulates the transformation of substrate (blue oval) to product (yellow triangle) at the catalytic rate ($k_{cat}$, green arrow). In allokairy, binding of substrate in the active site shifts the enzyme population from an inactive form to an active form, at an exchange rate ($k_{ex}$) that is similar to $k_{cat}$, resulting in time-dependent activity profiles (gradient of light green to dark green profiles). Following exhaustion of substrate, the enzyme population returns to the original equilibrium position. In allomorphy, reaction of the activating substrate, termed here allomorphic full activator (green hexagon), in the active site shifts the enzyme population from an inactive form to an active form, which stimulates the transformation of the native substrate (blue oval) to product (yellow triangle) at the maximal catalytic rate ($k_{cat}$, green arrow and green profile). However, reaction of alternatives substrates, termed here allomorphic partial activators (pink pentagon), in the active site are unable to shift the enzyme population from an inactive form to an active form, resulting in a slower overall catalytic rate ($k'_{cat}$, pink arrow and pink profile). The exchange rate ($k_{ex}$) between the two enzyme forms is much slower than $k_{cat}$. Following exhaustion of the allomorphic activator, the enzyme population returns to the original equilibrium position.

in all other substrate-free and TSA complex structures reported for βPGM$_{WT}$. The removal of this positively charged amine group from the active site of conformer B disrupts the charge balance in the vicinity of the D8 nucleophile and therefore transition state stability will be severely impaired[47,48]. However, kinetics data for the mutase reaction indicate that βPGM$_{P146A}$ is only ~20-fold less active than βPGM$_{WT}$, and has a similar affinity for βG1P. In the crystal structure of the βPGM$_{P146A}$:MgF$_3$:G6P TSA complex, which mimics the transition state of the phosphoryl transfer step, the K145–A146 peptide bond adopts the *cis*-isomer and the sidechain of K145 is able to engage in the active site. Hence, conformer A remains the preferred route for phosphoryl transfer in βPGM$_{P146A}$, despite it being ~1000 times less stable than in βPGM$_{WT}$ because of the *trans* to *cis* isomerisation of the K145–A146 peptide bond[46].

This disruption of the active site has regulatory significance, as conformer B of βPGM$_{WT}$ constitutes a substantial population of the resting substrate-free enzyme. In vitro, linear initial kinetics and full activation of the enzyme are achieved only when βG16BP is used as the phosphorylating agent. Phosphorylation of both βPGM$_{WT}$ and βPGM$_{P146A}$ with either F16BP or AcP leads to a lag phase, with the lag caused by F16BP lasting ~3.5 times longer than that observed with AcP. Moreover, real-time NMR experiments establish that the phosphorylation of both βPGM$_{WT}$ and βPGM$_{P146A}$ with either F16BP or AcP result in the generation of B$^P$. In βPGM$_{WT}$, B$^P$ isomerises completely into A$^P$ in <5 min,

whereas in βPGM$_{P146A}$, B$^P$ is consistently more populated than A$^P$. In contrast, B$^P$ is not observed for either βPGM$_{WT}$ or βPGM$_{P146A}$ when βG16BP is used as the phosphorylating agent. This result demonstrates that phosphorylation with βG16BP leads to the stabilisation of conformer A, resulting in production of A$^P$, regardless of the initial βPGM conformation, even for the βPGM$_{P146A}$ variant despite the considerable energetic cost of the conformer B to conformer A transition. Thus, conformer B and

$B^P$ are significantly less-active forms of βPGM, and the slow transition from $B^P$ to $A^P$ is part of the characteristic lag phase observed in the coupled assay kinetics. The longer lag in the F16BP experiments and the observation of a βPGM$_{WT}$:F16BP complex are consistent with F16BP having a slower phosphorylation rate than AcP. Therefore, βPGM is able to follow alternative kinetic pathways depending on the phosphorylating agent present, with its overall catalytic rate determined by the rate-limiting step in each pathway (Fig. 6a, b).

The response of βPGM to different phosphorylating agents also has functional significance. In *L. lactis*, the sole source of βG16BP is βPGM itself. In contrast, F16BP accumulates during glycolysis, reaching a concentration of up to ~50 mM[17]. Hence, F16BP is likely to be the primary source of βPGM activation in vivo, with AcP contributing to a lesser extent[43,44]. Correspondingly, a proportion of βPGM will populate the less-active $B^P$ species. Following a switch from glucose to maltose or trehalose metabolism, which will greatly increase the βG1P concentration (Supplementary Fig. 1), βG16BP will begin to accumulate (Fig. 1, 6a, b). As a result, the conformer B and $B^P$ populations will be recruited into the more active $A^P$ species. This two-state control mechanism allows *L. lactis* to effectively catabolise maltose or trehalose, while the increase in transcription of *pgmB* is in progress[20]. Furthermore, when carbohydrate levels are low, a significant proportion of the basal βPGM population will be maintained as conformer B, which will consequently reduce the undesirable dephosphorylation of F16BP and also hinder the conversion of G6P to βG1P.

The multi-second substrate-dependent non-allosteric conformational exchange mediated through *cis–trans* proline isomerisation seen in βPGM represents a mechanism of post-translational enzyme regulation. This regulation mechanism relies on the existence of alternative pathways with different rate-limiting steps, where the catalytic rate depends on the capacity of an activator, acting as a substrate, to bias the enzyme population towards the fastest pathway, by stabilising the most active conformation (Fig. 7). Similar to allostery and allokairy, this mechanism depends on the ability of the enzyme to adopt at least two conformations with distinct activities, but does not require the binding of an additional effector to the protein, nor an equivalence between the conformational exchange rate and the catalytic rate. We suggest the name allomorphy for this mechanism, from the greek *allos* meaning other and *morphe* meaning form, in keeping with the previously described allostery and allokairy mechanisms[15].

Allomorphy may modulate the activity of other monomeric enzymes with hysteretic behaviour, i.e., those that exhibit a burst or lag phase in their kinetic profile[49]. Several theoretical models have been put forward to rationalise hysteretic behaviour, such as the mnemonic[50] and the ligand-induced slow transition[51] models, but detailed structural-based molecular mechanisms have proved elusive. To our knowledge, only one such mechanism, allokairy in human glucokinase, has been described in detail[15,16]. Allomorphy is a different fine control regulatory mechanism and is potentially widespread, at least across phosphomutases; for example, both rabbit muscle and *L. lactis* α-phosphoglucomutases appear to be hysteretic enzymes[52,53], but belong to very different protein superfamilies. Like βPGM, these enzymes require a phosphorylating agent to initiate the catalytic cycle and, for the latter, the use of the reaction intermediate results in linear kinetics, whereas alternative phosphorylating agents produce a lag phase in their kinetic profiles. Similarly, α-phosphomannomutase from *Galdieria sulphuraria*, which also requires the addition of a phosphorylating agent to initiate the catalytic cycle, exhibits linear kinetics when α-mannose 1-phosphate and α-mannose 1,6-bisphosphate (or α-glucose 1-phosphate and αG16BP) are

included in the reaction, but has a lag phase when there is a mismatch between substrate and phosphorylating agent, or when F16BP is used as the phosphorylating agent[54]. All of these observations are consistent with the presence of allomorphic control.

In summary, allomorphy is a fine control mechanism by which part of an enzyme population is maintained in a more latent state, and is quickly switchable between high and low activity levels, without allosteric effectors. It delivers important control with which *L. lactis* is able to reconcile two seemingly contradictory requirements: the need to maximise its responsiveness to changes in carbohydrate source and the need to minimise unproductive diversion of valuable metabolites.

## Methods

**β-phosphoglucomutase (βPGM) expression and purification**. Wild-type βPGM (βPGM$_{WT}$) and the P146A variant (βPGM$_{P146A}$) proteins were expressed using either $^{15}$N or $^{2}$H$^{15}$N$^{13}$C isotopic enrichment[55] and purified using methodology that minimised the presence of contaminating phosphoryl transfer enzymes (e.g., phosphoglucose isomerase and βPGM from *E. coli*)[31,42]. The βPGM$_{WT}$ and βPGM$_{P146A}$ genes were cloned into the pET-22b(+) plasmid, transformed into *E. coli* strain BL21(DE3) cells and expressed in defined isotopically labelled minimal media. Cells were grown at 37 °C with shaking until OD$_{600nm}$ = 0.6, then cooled at 25 °C and induced with 0.5 mM isopropyl β-D-1-thiogalactopyranoside for a further 18 h. Cells were harvested by centrifugation at 10,000 rpm for 10 min. The cell pellet was resuspended in ice-cold lysis buffer (50 mM K$^{+}$ HEPES (pH 7.2), 2 mM NaN$_3$, 1 mM EDTA) supplemented with cOmplete™ protease inhibitor cocktail and lysed by six cycles of sonication. The cell lysate was cleared by centrifugation at 20,000 rpm for 35 min at 4 °C. The supernatant was filtered using a 0.22 μm syringe filter and loaded onto a DEAE-Sepharose fast flow anion-exchange column. Proteins bound to the column were eluted with a gradient of 0–50% lysis buffer containing 1 M NaCl. Fractions containing βPGM were purified further using a Hiload 26/600 Superdex 75 size-exclusion column previously equilibrated with lysis buffer containing 1 M NaCl. Final fractions were pooled, buffer exchanged into 50 mM K$^{+}$ HEPES buffer (pH 7.2) containing 2 mM NaN$_3$ and concentrated to ~1.6 mM for storage at −20 °C.

**Reagents**. Unless otherwise stated, reagents were purchased from Sigma-Aldrich, GE Healthcare, Melford Laboratories or CortecNet. βG1P was synthesised enzymatically from maltose using maltose phosphorylase (EC 2.4.1.8)[31]. A solution of maltose (600 mM) was incubated overnight with 1.2 units mL$^{-1}$ maltose phosphorylase in 0.5 M sodium phosphate buffer (pH 7.0) at 30 °C and βG1P production was confirmed using $^{31}$P NMR spectroscopy. Maltose phosphorylase was removed using a Vivaspin (5 kDa MWCO) and the resulting flow-through was used without further purification. βG16BP was produced enzymatically from βG1P and AcP using the D170N variant of βPGM (βPGM$_{D170N}$; expressed and purified as detailed above)[56]. βG1P and AcP were incubated with βPGM$_{D170N}$ for 4 h at 25 °C and the reaction was quenched by heating at 90 °C for 10 min. βG16BP was purified using barium salt precipitation.

**NMR spectroscopy**. $^{1}$H$^{15}$N-TROSY NMR spectra of βPGM$_{WT}$ and βPGM$_{P146A}$ were acquired at 298 K using 0.5–1 mM $^{15}$N-βPGM in standard NMR buffer (50 mM K$^{+}$ HEPES (pH 7.2), 5 mM MgCl$_2$, 2 mM NaN$_3$ with 10% (v/v) $^{2}$H$_2$O and 1 mM TSP). Typically, $^{1}$H$^{15}$N-TROSY spectra were accumulations of 16 transients, with 256 increments and spectral widths of 32–36 ppm centred at 120 ppm in the indirect $^{15}$N dimension. $^{1}$H$^{15}$N-TROSY-based ZZ-exchange experiments were performed at mixing times of 100, 300, 500 and 900 ms. Rapid acquisition $^{1}$H$^{15}$N BEST-TROSY experiments to monitor the steady-state behaviour of $^{15}$N-βPGM$_{WT}$ (0.2 mM) and $^{15}$N-βPGM$_{P146A}$ (0.2 mM) were acquired in standard kinetic buffer (200 mM K$^{+}$ HEPES (pH 7.2), 5 mM MgCl$_2$, 2 mM NaN$_3$ with 10% (v/v) $^{2}$H$_2$O and 1 mM TSP) containing either 50–100 mM F16BP, 60–100 mM AcP or 35 mM βG16BP. The $^{1}$H$^{15}$N BEST-TROSY spectra were recorded at 298 K using a Bruker 600 MHz Neo spectrometer equipped with a 5-mm TCI cryoprobe and z axis gradients as 11 min experiments (16 transients, 128 increments and a recycle delay of 0.2 s) with selective $^{1}$H pulses centred on the amide region (8.3 ppm). Excitation pulses (90°) were 1.7 ms (pulse shape Eburp2), whereas refocusing pulses (180°) were 1.4 ms (pulse shape Reburp). The experimental dead-time was ~5 min.

For βPGM$_{P146A}$ prepared in standard kinetic buffer containing 50 mM F16BP, $B^P$ dephosphorylation was monitored at 298 K by consecutive one-dimensional $^{1}$H NMR experiments recorded with 16 transients, a 1 s recycle delay and a spectral width of 32 ppm centred on the water signal. Following 0.3 Hz Lorentzian apodisation and baseline correction, normalised integral values of the F16BP peak (4.22–4.15 ppm) were plotted against time to give a kinetic profile. The initial linear steady-state portion of the kinetic profile was fitted using a linear least-squares fitting algorithm included in MATLAB 2018a to derive an apparent dephosphorylation rate constant.

To observe the species present immediately following the addition of phosphorylating agent to βPGM_{WT}, NMR experiments were recorded with the use of a homemade rapid mixing device. The equipment comprised a 2 m length of 0.8 mm internal diameter EFTE tubing (GE Healthcare), connected at one end to a 1 mL syringe and inserted at the other end through the lid of an NMR tube. The tubing was loaded with phosphorylating agent (550 μL 100 mM F16BP or 250 μL 320 mM AcP, prepared in standard kinetic buffer) and a small, additional volume of air was drawn in to prevent premature mixing of the phosphorylating agent with the 550 μL 1.2 mM $^{15}$N-βPGM_{WT} sample prepared in standard kinetic buffer. The rapid mixing device was loaded into a Bruker 600 MHz Neo spectrometer and allowed to equilibrate thermally at 298 K. Following mixing by syringe action of the phosphorylating agent (final concentration: 50 mM F16BP or 100 mM AcP) with the βPGM_{WT} sample, the spectrometer was locked (with ~7% (v/v) $^2$H$_2$O), tuned and shimmed, and the $^1$H 90° pulse length was calibrated. A series of $^1$H$^{15}$N BEST-TROSY spectra were recorded as 142 s experiments (4 transients, 128 increments and a recycle delay of 0.15 s). The experimental dead-time was ~156 s.

Multi-dimensional heteronuclear NMR spectra for $^1$H, $^{15}$N and $^{13}$C backbone resonance assignment of $^2$H$^{15}$N$^{13}$C-βPGM_{WT} in standard NMR buffer containing 10 mM tris were acquired at 298 K on a Bruker 800 MHz Avance III spectrometer equipped with a 5-mm TCI cryoprobe and $z$ axis gradients. The standard Bruker suite of $^1$H$^{15}$N-TROSY and 3D TROSY-based constant time experiments were acquired (HNCO, HN(CA)CO, HNCA, HN(CO)CA, HNCACB, HN(CO)CACB) using non-uniform sampling (NUS) with a multi-dimensional Poisson Gap scheduling strategy with exponential weighting[57]. NUS data were reconstructed using multi-dimensional decomposition in TopSpin3[58]. Backbone resonance assignments for conformer A and conformer B present simultaneously in the spectra were obtained using a simulated annealing algorithm employed by the asstools assignment program[55] and assignments were confirmed using sequential backbone amide to amide correlations obtained from TROSY-based (H)N(COCA)NNH and H(NCOCA)NNH experiments[59]. Multi-dimensional heteronuclear NMR spectra for $^1$H, $^{15}$N and $^{13}$C backbone resonance assignment of the $^2$H$^{15}$N$^{13}$C-βPGM_{P146A}:MgF$_3$:G6P TSA complex in standard NMR buffer containing 15 mM NaF and 10 mM G6P were acquired at 298 K on a Bruker 800 MHz Avance I spectrometer equipped with a 5-mm TXI probe and $z$ axis gradients. $^1$H$^{15}$N-TROSY and 3D TROSY-based constant time experiments were acquired (HN(CA)CO and HNCACB) and backbone resonance assignments were obtained using asstools[55]. Multi-dimensional heteronuclear NMR spectra for $^1$H, $^{15}$N and $^{13}$C backbone resonance assignment of phosphorylated $^2$H$^{15}$N$^{13}$C-βPGM_{P146A} in standard kinetic buffer containing 75–100 mM F16BP were acquired at 298 K on a Bruker 800 MHz Neo spectrometer equipped with a 5-mm TXI probe and $z$ axis gradients. $^1$H$^{15}$N-TROSY and 3D TROSY-based constant time experiments were acquired (HNCO and HNCA) using NUS with a multi-dimensional Poisson Gap scheduling strategy with exponential weighting[57]. NUS data were reconstructed using multi-dimensional decomposition in TopSpin4[58]. TROSY resonances were assigned by comparing the correlated $^{13}$C chemical shifts with those of βPGM_{P146A} (BMRB 27920[42]) and the βPGM_{WT}:BeF$_3$ complex (BMRB 17851[35]). Experiments were processed using TopSpin (Bruker) or FELIX (Felix NMR, Inc.) and NMR figures were prepared using FELIX. $^1$H chemical shifts were referenced relative to the internal TSP signal resonating at 0.0 ppm, and $^{13}$C and $^{15}$N chemical shifts were referenced indirectly using nuclei-specific gyromagnetic ratios. Differences in chemical shift were calculated as: $\Delta\delta = [(\delta_{HN-X} - \delta_{HN-Y})^2 + (0.12 \times (\delta_{N-X} - \delta_{N-Y}))^2]^{1/2}$, where $X$ and $Y$ are the two species being compared.

Reaction kinetics for βPGM_{P146A}-catalysed reactions were followed using a Bruker 500 MHz Avance DRX spectrometer (operating at 202.456 MHz for $^{31}$P) equipped with a room-temperature broadband probe. The equilibration of 10 mM βG1P with G6P by 1–3 μM βPGM_{P146A} was measured in standard kinetic buffer at 298 K. The reaction was initiated by and timed from the addition of 20 mM AcP and monitored by the acquisition of consecutive $^{31}$P spectra without proton-phosphorus decoupling and a 1 s recycle delay. A spectral width of 50 ppm centred at −10 ppm enabled the observation of the relevant phosphorus signals. Normalised integral values of both the βG1P and G6P peaks following baseline correction and 5 Hz Lorentzian apodisation were plotted against time to give kinetic profiles. The linear steady-state portion of the G6P integral data was fitted using a linear least-squares fitting algorithm to derive the observed rate constant, $k_{obs}$.

One-dimensional $^{19}$F NMR experiments were acquired at 298 K on a Bruker 500 MHz Avance III spectrometer (operating at 470.536 MHz for $^{19}$F) equipped with a 5-mm QCI-F cryoprobe and $z$ axis gradients. Samples were prepared using 0.5 mM $^{15}$N-βPGM_{WT} or $^{15}$N-βPGM_{P146A} in standard NMR buffer (without 10% (v/v) $^2$H$_2$O) containing 15 mM NaF and 10 mM G6P. The spectrometer lock was provided by $^2$H$_2$O sealed inside a glass capillary tube inserted into the sample tube. Typically, ~6000 transients were acquired without $^1$H decoupling over a spectral width of 120 ppm and were processed with sinebell functions shifted by 60°.

For NMR experiments conducted in filtered milk, fresh skimmed bovine milk purchased from a local supermarket was filtered using a 10 kDa MWCO Vivaspin (Sartorius) to remove fat micelles and milk proteins. A 1 mM $^{15}$N-βPGM_{WT} sample prepared in standard NMR buffer was diluted fivefold with the resulting milk flow-through and 10% (v/v) $^2$H$_2$O and 1 mM TSP were added to the sample. Experiments were acquired at 298 K on a Bruker 800 MHz Avance I spectrometer

equipped with a 5-mm TXI probe and $z$ axis gradients. The pH of the sample was estimated as pH 6.8 using the residual $^1$H resonances of HEPES buffer originating from the standard NMR buffer.

For NMR experiments conducted in the absence of salt, a 1.4 mM $^{15}$N-βPGM_{WT} sample in standard NMR buffer containing an additional 200 mM NaCl was buffer-exchanged into freshly produced deionised water (18.2 MΩ.cm, Purelab Classic, Elga-Veolia), with an equivalent dilution of the previous buffer by a factor of $18.7 \times 10^6$. The resulting sample contained 0.9 mM βPGM_{WT} at pH 6.3 and was supplemented with 10% (v/v) $^2$H$_2$O for the deuterium lock. Experiments were acquired at 298 K on a Bruker 800 MHz Avance I spectrometer equipped with a 5-mm TXI probe and $z$ axis gradients.

**Kinetic experiments using coupled assays.** All kinetic assays for βPGM_{WT} and βPGM_{P146A} were conducted at 298 K using a FLUOstar OMEGA microplate reader and the BMG LABTECH Reader Control Software (version 5.11) (BMG Labtech) in standard kinetic buffer (200 mM K$^+$ HEPES (pH 7.2), 5 mM MgCl$_2$, 1 mM NaN$_3$) in a 160 μL reaction volume. The rate of G6P production was measured indirectly using a glucose 6-phosphate dehydrogenase (G6PDH) coupled assay, in which G6P is oxidised and concomitant NAD$^+$ reduction is monitored by the increase in absorbance at 340 nm (NADH extinction coefficient = 6220 M$^{-1}$ cm$^{-1}$). βPGM_{WT} and βPGM_{P146A} concentrations were determined using a NanoDrop OneC spectrophotometer (Thermo Scientific) and diluted accordingly (βPGM extinction coefficient = 19,940 M$^{-1}$ cm$^{-1}$). For the determination of $k_{cat}$ and $K_m$ values for βPGM_{WT}, the reaction was initiated by dilution of the enzyme prepared in standard kinetic buffer to a final concentration of 1 nM βPGM_{WT} in solutions of 1 mM NAD$^+$ and 5 units mL$^{-1}$ G6PDH and variable concentrations of βG1P (10, 20, 30, 50, 70, 100, 150, 200, 300, 500, 700 μM) and βG16BP (0.4, 1, 2, 5, 10 μM). For the determination of $k_{cat}$ and $K_m$ values for βPGM_{P146A}, the reaction was initiated by dilution of the enzyme prepared in standard kinetic buffer to a final concentration of 100 nM βPGM_{P146A} in solutions of 1 mM NAD$^+$ and 5 units mL$^{-1}$ G6PDH and variable concentrations of βG1P (5, 10, 15, 20, 30, 50, 70, 100, 200, 300, 500 μM) and βG16BP (2, 5, 10, 35, 50, 100 μM). The initial rate of G6P production was fitted using a linear least-squares fitting algorithm to determine the reaction velocity ($v_0$) at each βG1P and βG16BP concentration at a total enzyme concentration ($E_T$). Mean data from triplicate measurements were subsequently globally fitted to Eq. 1[28], which is derived for a ping–pong mechanism and adapted to account for βG1P inhibition ($K_i$) to calculate $k_{cat}$ and individual $K_m$ values ($K_{βG1P}$ and $K_{βG16BP}$), with their corresponding standard deviations, using an in-house python non-linear least-squares fitting program.

$$v_0 = \frac{k_{cat}[E_T][\beta G1P][\beta G16BP]}{[\beta G1P][\beta G16BP] + K_{\beta G1P}[\beta G16BP] + K_{\beta G16BP}[\beta G1P]\left(\frac{K_i + [\beta G1P]}{K_i}\right)} \quad (1)$$

Kinetic experiments demonstrating the effect of different phosphorylating agents were conducted by the addition of either 5 nM βPGM_{WT} or 200 nM βPGM_{P146A} to solutions containing either 1 mM F16BP, 8 mM AcP or 10 μM βG16BP, together with 1 mM NAD$^+$, 5 units mL$^{-1}$ G6PDH and 50 μM βG1P. F16BP represents an equilibrium mixture of an α-anomer (15%), a β-anomer (81%) and two open chain forms with an interconversion rate of 8 s$^{-1}$[60].

**X-ray crystallography.** For the crystallisation experiments of βPGM_{WT}, βPGM_{P146A}, the βPGM_{P146A}:MgF$_3$:G6P TSA complex and the βPGM_{WT}:citrate complex, frozen aliquots of βPGM_{WT} or βPGM_{P146A} in standard native buffer (50 mM K$^+$ HEPES (pH 7.2), 5 mM MgCl$_2$, 2 mM NaN$_3$) were thawed on ice and centrifuged briefly to pellet insoluble material. For the βPGM_{P146A}:MgF$_3$:G6P TSA complex, 15 mM NaF and 10 mM G6P were added to the βPGM_{P146A} sample, whereas for the βPGM_{WT}:citrate complex, 50 mM citrate was added to the βPGM_{WT} sample. Solutions were adjusted to a final protein concentration of 0.4–0.6 mM, incubated for 1 h and mixed 1:1 with precipitant (24–34% (w/v) PEG 4000, 200 mM sodium acetate and 100 mM tris-HCl (pH 7.5)). Crystals were grown at 290 K by hanging-drop vapour diffusion using a 2 μl drop suspended on a siliconised glass cover slip above a 700 μL well. Rod-shaped or large plate crystals grew typically over several days. Crystals were harvested using a mounted Litho-Loop (Molecular Dimensions Ltd) and were cryo-protected in their mother liquor containing an additional 25% (v/v) ethylene glycol (and 50 mM citrate for the βPGM_{WT}:citrate crystals) prior to plunging into liquid nitrogen. Diffraction data were collected at 100 K on the MX beamlines at the Diamond Light Source (DLS), Oxfordshire, United Kingdom.

Data were processed using the xia2 pipeline[61] and resolution cutoffs were applied using CC-half values and Aimless[62]. The crystals diffracted in the P2$_1$2$_1$2$_1$ spacegroup, with cell dimensions that varied depending on the degree of enzyme closure. Structures were determined by molecular replacement with MolRep (version 11)[63] using the highest resolution model with the most appropriate cap and core domain relationship as a search model. Model building was carried out in COOT (version 0.8.8)[64] with ligands not included until the final rounds of refinement with REFMAC5[65] so that they could be built into unbiased difference Fourier maps. The βPGM_{P146A}:MgF$_3$:G6P TSA complex structure was refined with anisotropic B-factors, whereas both the βPGM_{WT} structures and the substrate-free βPGM_{P146A} structure were refined isotropically. Structure validation was carried out in COOT and MolProbity (version 4.4)[66]. Superpositions and crystallographic

figures were prepared using PyMOL (The PyMOL Molecular Graphics System, version 1.8, Schrödinger, LLC). To confirm the isomerisation state of the K145–A146 peptide bond in the structures of substrate-free βPGM$_{P146A}$ and the βPGM$_{P146A}$:MgF$_3$:G6P TSA complex, difference density maps (Fo–Fc) were generated using REFMAC5 with the S144–P148 segment omitted from the final structures. Omit map figures were prepared using CCP4mg (version 2.10.9)[67]. Additional details for X-ray crystallography data collection, data processing and refinement are provided in Supplementary Table 1.

## Data availability

Data supporting the findings of this manuscript are available from the corresponding author upon reasonable request. The atomic coordinates and structure factors have been deposited in the Protein Data Bank (www.rcsb.org) with the following codes: βPGM$_{WT}$:citrate complex (PDB 6YDM), substrate-free βPGM$_{WT}$ (PDB 6YDL), substrate-free βPGM$_{P146A}$ (PDB 6YDK) and βPGM$_{P146A}$:MgF$_3$:G6P TSA complex (PDB 6YDJ). The NMR chemical shifts have been deposited in the BioMagResBank (www.bmrb.wisc.edu) with the following accession numbers: substrate-free βPGM$_{WT}$ conformer A (BMRB 28095), substrate-free βPGM$_{WT}$ conformer B (BMRB 28096) and βPGM$_{P146A}$:MgF$_3$:G6P TSA complex (BMRB 28097).

## Code availability

Code developed in Python3 and bash for this study is publicly available under an MIT license and can be found on GitHub [https://doi.org/10.5281/zenodo.4022248].

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

## Acknowledgements

We thank Dr Claudine Bisson for helpful discussions and interpretation of X-ray crystallography data. We thank the beamline scientists at the DLS for the provision of synchrotron radiation facilities and assistance with data collection. This research was supported by the Biotechnology and Biological Sciences Research Council (BBSRC; H.P.W.—Grant Number X/009906-20-26, N.J.B.—Grant Number BB/M021637/1 and BB/S007965/1, C.R.T.—Grant Number BB/P007066/1), Consejo Nacional de Ciencia y Tecnologia, Mexico (CONACYT; F.A.C.N.—Grant Number 472448) and The Royal Society (S.R.D.—Grant Number R/152968).

## Author contributions

H.P.W., F.A.C.N., N.J.B., C.R.T., A.J.R., J.P.W. designed research; H.P.W., F.A.C.N., C.R.T., A.J.R. produced isotopically enriched protein; H.P.W. performed coupled assay kinetic experiments; H.P.W., C.R.T. produced and purified βG1P and βG16BP; H.P.W., F.A.C.N., C.R.T., A.M.H., M.J.C. acquired NMR experiments; H.P.W., F.A.C.N., N.J.B., C.R.T. analysed NMR data; F.A.C.N., N.J.B., C.R.T. performed backbone resonance assignments; H.P.W., A.J.R., S.R.D. performed and analysed X-ray crystallography experiments; F.A.C.N. conceived and developed the allomorphy mechanism; H.P.W., F.A.C.N., N.J.B., C.R.T., J.P.W. wrote the paper with help from all authors.

## Competing interests

The authors declare no competing interest.

## Additional information

**Peer review information** *Nature Communications* thanks Andrea Mattevi and other, anonymous, reviewers for their contributions to the peer review reports of this work. Peer review reports are available.

