## [Peer Review File · Nature Communications]

REVIEWER COMMENTS

Reviewer #1 (Remarks to the Author):

This is an excellent paper reporting very thorough work to elucidate the mechanism of activation in beta-phosphoglucosyltransferase (bPGM). It follows a series of impressive papers by the Waltho group that has provided an increasing level of detail on the catalytic mechanism of bPGM. Here they use a combination of multinuclear NMR, X-ray crystallography and kinetic activity experiments. The authors describe a novel mode of enzyme regulation where different phosphorylating agents activate the enzyme to different degrees, thereby enabling rapid switching between higher and lower activity in response to changing substrate conditions and metabolic status of the cell. This important paper will have a broad impact in the biomolecular sciences.

The current paper relies on careful interpretation of NMR spectra to identify specific conformers, phosphorylated states and complexes, which is made possible by comparison to spectra of previously characterized complexes, such as model systems of the substrate bound ground-state and the transition state.

I recommend that the authors address a couple of issues that I find slightly confusing or otherwise need clarification:

1. In the presence of bG16BP, you observe a single peak from A113 attributed to the A:bG16BP complex (i.e., the non-phosphorylated cis-conformer bound to bG16BP). This peak has weak intensity, suggesting that bPGM undergoes conformational exchange, which leads to the question whether you can confidently say that other states are not present at lower, but still significant, populations than that of the detected species. Would it be possible to determine the relative populations of phosphorylated cis and trans forms using ^{31}P NMR?

Related to this issue, you state in the abstract and elsewhere: "...b-glucose 1,6-bisphosphate, ..., phosphorylates both conformers into the more active species." It is not clear to me why you believe that both the A (cis) and B (trans) conformers become phosphorylated by bG16BP. If this is the case, then why does the kinetics not show a lag phase similar to that observed with F16BP? It seems to me that a more likely explanation would be that bG16BP can phosphorylate only the A conformer. It appears (but see above) that bG16BP binds with strong preference to the A conformer. Please clarify this issue.

Continuing on this thread, would it be possible to determine the crystal structure of the D10N mutant in complex with F16BP? Perhaps this structure would reveal both cis and trans conformers in the complex. Perhaps modeling can reveal any differences in the binding modes of bG16BP and F16BP?

2. In Fig. 5b, the kinetic profiles of the P146A mutant with bG16BP and AcP appear to be biphasic. Please comment on this.

3. Did I get it right that the crystal structure of 'apo' bPGM shows only the cis conformer? Why is this? Can the observation be explained by crystal contacts? Please comment on this.

Reviewer #2 (Remarks to the Author):

The manuscript by Wood et al. describes NMR-based, crystallographic and kinetic evaluation of the lag phase in the activity of the enzyme geta-phosphoglucomutase (β PGM) from *L. lactus*. The interesting facet here is how an enzyme which is monomeric might be regulated without the need for an additional binding site or differential activator but by the substrate itself in the phosphorylation step of the mutase reaction. That is, the enzyme undergoes a conformational change (in this case trans to cis) which is better facilitated by some phosphorylating agents than others. Because there is no one method that shows that the structure observed is connected to function there are extensive experiments to tie together the results from structure and kinetics. The technical execution of the work is very good (although see some questions about kinetics and some problems with description of structures).

The main question in terms of potential impact of the findings comes from the main point of the manuscript which is the observation of what the authors dub allomorphy. Other work has described the the adoption of discrete functional states that correlate with the enzyme's major conformational states and are redistributed in the presence of a regulatory effector (see Nikos S. Hatzakis, et al. (2012) Single Enzyme Studies Reveal the Existence of Discrete Functional States for Monomeric Enzymes and How They Are "Selected" upon Allosteric Regulation. *Journal of the American Chemical Society* 134 (22), 9296-9302 DOI: 10.1021/ja3011429). Another factor here is that the phosphorylating activator is the intermediate of the enzyme reaction, thus it is not completely unexpected to find that it has the most stabilizing or organizing effect on the enzyme and leads to population of the most active conformer.

The authors may wish to consider the following comments (given in order of appearance; page numbers from pdf file for review).

P 7 line 155- "While β G16BP is able to phosphorylate both conformers into the more active species, alternative phosphorylation agents such as AcP and F16BP generate phosphorylated forms of both conformers, resulting in lower β PGM activity." This statement - as a stand alone makes no sense- reword to distinguish what is meant by the two possibilities- they seem the same

Page: 9- The authors state "The β PGMWT conformers are present in the spectra as a result of slow conformational exchange rather than as chemically distinct species, since the addition of 3 mM BeCl_2 and 10 mM NH_4F to the β PGMWT sample induced the population of a single β PGMWT: BeF_3 complex"

Please clarify- is this conformer A or conformer B?

Page: 10- in the description of the crystal structure "Two monomeric chains are present in the asymmetric unit, both of which overlaid closely with a previously reported substrate-free β PGMWT structure (PDB 2WHE29: non-H atom RMSDs of 0.56 Å and 0.95 Å), thus both chains reflect the binding of citrate to only one of conformer A or conformer B." This is very confusing- how can both chains reflect the binding of citrate to one conformer. Is citrate bound to one or both chains? This should be clarified. Also if citrate is bound to one or both- it means there is no change in structure as rmsd is small compared to unbound for both chains. How then is one chain designated as conformer A and the other as B? Are the authors saying that one conformer of the crystal structure is consistent with one set of NMR assignments (conformer A) and the other with conformer B ie – one is cis and the other trans? Please clarify.

Page: 11 – The authors state after description of the NMR and the X-ray crystal structures “these experiments show that both conformer A and conformer B remain well-populated under near-physiological conditions.- the X-ray structures are performed under very different conditions than NMR (high salt and precipitant) so it is not apparent how both these experiments are under physiological conditions?

It is notable that the ^{13}C chemical shift for P146 corroborates the presence of a cis K145-P146 peptide bond in solution for conformer A. However for conformer B, the absence of proline ^{13}C resonances for P146 precluded the identification of the trans isomerisation state for this proline. Thus the evidence for the trans conformer is indirect. However, the crystal structure together with the NMR results are consistent with the bPGMP146A structure being a close model of conformer B ie. the trans conformer

Page: 13 The authors state "The D137-A147 loop also shows significant positional variation between different substrate-free β PGMWT structures (2WHE and 1ZOL... implying that conformer B is present in some β PGMWT crystal structures" but this is not precisely true - although the start of this segment (137-143) varies more between these two structures- after residue Ser144 they are nearly identical such that 144, 145 146 and 147 overlay very nicely. Although the NMR data is more convincing, the authors need to have better arguments to support their model that conformer B is present in some unliganded WT crystal structures

The authors state "In β PGMWT, the e-amino group of K145 is coordinated by the carbonyl oxygen atom of A113, the carboxylate side chain of E169, and a water molecule that is replaced in the transition state analogue (TSA) complex (PDB 2WF529) by a fluoride ion that mimics an oxygen atom of the transferring phosphoryl group." But looking at the structure - it is also making a bond to the catalytically essential Asp 8. This bond is consistent between the unliganded and liganded TSA forms and should be noted here.

Page: 15- The authors calculate the kinetics for the P146A variant and find "Initial rate measurements were made at several bG1P and bG16BP concentrations (10-500 μM and 0.4-100 μM respectively) and were globally fit to the equation used for β PGMWT above. The fitting yielded values for k_{cat} of $19.2 \pm 0.2 \text{ s}^{-1}$, K_{m} (β G1P) of $157 \pm 3 \mu\text{M}$, $337 K_{\text{m}}$ (β G16BP) of $175 \pm 3 \mu\text{M}$ for β PGMP146A" A possible problem here is that in this series of experiments 10 μM beta G16BP was used as phosphorylating agent but that is ten fold below the calculated K_{m} of 157 so little of the enzyme may be phosphorylated simply due to binding of the agent.

Page: 16- The authors state "Additionally, the increase in K_{m} for bG16BP in β PGMP146A reflects the energetic cost of the trans to cis isomerisation of the K145-A146 peptide bond⁴⁶." This may be true if K_{m} represents a dissociation constant for bG16BP, but as K_{m} also includes rate constant(s), then the difference in K_{m} cannot be equated with this energy. If the authors have an argument to show that K_{m} equals K_{d} then it should be given here.

Page: 17- The rate constant for dephosphorylation (through hydrolysis) was determined to be $0.02 \pm 0.002 \text{ s}^{-1}$ from the rate of reduction of the free F16BP concentration in ^1H NMR experiments.- is this the correct statement?- it seems that the rate of disappearance of the F16BP would equate with the rate of enzyme phosphorylation not of dephosphorylation. When the phosphoryl group is removed from F16BP the enzyme is phosphorylated and dephosphorylation ensues from the hydrolysis of the

enzyme

Page: 22- The multi-second conformational exchange due to cis-trans proline isomerisation seen in the substrate-free form of β PGM represents a hitherto unidentified mechanism of post-translational enzyme regulation. As described in this reviewers opening statements, I believe this mechanism has been previously observed for monomeric enzymes.

ref 10 and 32 are the same

Reviewer #3 (Remarks to the Author):

This manuscript describes an extensive experimental investigation into β -phosphoglucomutase, which results in the proposal of a novel regulatory mechanism that the authors call "allomorphy". NMR and x-ray crystallographic methods have been used to characterize two different conformers of β -PGM, resulting from isomerization of a peptide bond, and the authors offer convincing arguments that they serve a biological purpose. The enzyme has a short-lived phosphorylated form that is part of the catalytic cycle. A high activity enzyme form is stabilized when the phosphorylated enzyme is formed from the natural substrate; alternative phosphorylating agents result in the preponderant formation of a lower activity conformer.

Allomorphy is a novel concept, and the experimental data as interpreted by the authors provide solid support for it. It is a rather subtle concept, however, and moving Figure 7 to the introduction would better prepare the reader for the arguments that follow.

Figure 5b shows a significant slow phase following the initial rapid phase when the P146A enzyme is phosphorylated by acetyl-P or β -G16BP. Do the authors know what causes this?

Overall, this is a well-written manuscript, and the following are minor points.

-line 40. "...phosphorylates both conformers into the more active species." This is odd phrasing; a compound is phosphorylated, not phosphorylated *into* something.

-lines 110-114. Phosphorylases catalyze phosphorylation reactions, not hydrolysis by a P_i -dependent enzyme.

-Figure 6. What is the meaning of the double-headed arrows that connect BP and AP?

Peter Tipton

Reviewer #4 (Remarks to the Author):

Wood et al describe a very thorough analysis of the unusual regulatory mechanism in an enzyme, β -phosphoglucomutase, of the basic metabolism. This mutase functions as a phosphoenzyme. The phosphate group tends to easily hydrolyze. Therefore, the enzyme needs a continuous supply of phosphate donors that, physiologically, can be β -glucose 1,6-bisphosphate (a dissociable reaction intermediate) or fructose-1,6-bisphosphate. The authors demonstrate that these two compounds are not at all equivalent. β -Glucose 1,6-bisphosphate efficiently phosphorylate the enzyme and promotes its activate conformation. By contrast, fructose-1,6-bisphosphate less efficiently modifies the enzyme and hardly promotes its conversion to the active conformation. Indeed, the NMR, crystallographic, and mutagenesis studies described in this manuscript demonstrate the existence of

inactive and active states and interconversion between the two forms can limit the reaction and cause a lag phase. In other words, β -glucose 1,6-bisphosphate primes the enzyme conformation for catalysis whereas fructose-1,6-bisphosphate does not so. The conformational change involves the isomerization of a peptide bond. The idea that the substrate (or its analogue) can alter the equilibrium between two protein conformations is not new in enzymology; certain enzymes seem to keep a memory of substrate binding so that once they start to turn substrates, they increase their reactivity. The value of this manuscript is in the depth of the analysis and the implications of these findings for regulation of metabolism. Therefore, I think that the manuscript is a valuable contribution with some new insight into a "classical" problem of enzymology.

I have a few points:

-Line 212: this is a critical point. The authors use milk to mimic the physiological environment. Are we sure that this is indeed the case? Why should milk mimic the cytosol? This point is particularly important in light of the data reported in line 206: 200 mM NaCl trigger the active conformation. This salt concentration does not seem too far from a "physiological" ionic strength. This point needs to be discussed to strengthen the idea that the observed kinetic/regulatory effects can indeed be relevant for metabolism regulation in the cell.

-Another critical point concerns the electron density of Figure 4C, which is not convincing. As it is represented, one would guess that the peptide is in multiple conformations rather than in a single trans conformation.

-I hate to suggest additional experiments. But in this case, a simple experiment that could be done (e.g. with a nanotemper instrument) would be to measure the thermal stability of the WT and mutant proteins in their different states (native, phosphorylated, activated, transition-state like as used for crystallization etc). This would give an idea about the effect of the cis-trans isomerization on protein stability in the WT and the Pro mutant.

REVIEWER COMMENTS

Reviewer #1 (Remarks to the Author):

This is an excellent paper reporting very thorough work to elucidate the mechanism of activation in beta-phosphoglucosyltransferase (bPGM). It follows a series of impressive papers by the Waltho group that has provided an increasing level of detail on the catalytic mechanism of bPGM. Here they use a combination of multinuclear NMR, X-ray crystallography and kinetic activity experiments. The authors describe a novel mode of enzyme regulation where different phosphorylating agents activate the enzyme to different degrees, thereby enabling rapid switching between higher and lower activity in response to changing substrate conditions and metabolic status of the cell. This important paper will have a broad impact in the biomolecular sciences.

The current paper relies on careful interpretation of NMR spectra to identify specific conformers, phosphorylated states and complexes, which is made possible by comparison to spectra of previously characterized complexes, such as model systems of the substrate bound ground-state and the transition state.

I recommend that the authors address a couple of issues that I find slightly confusing or otherwise need clarification:

1. In the presence of bG16BP, you observe a single peak from A113 attributed to the A:bG16BP complex (i.e., the non-phosphorylated cis-conformer bound to bG16BP). This peak has weak intensity, suggesting that bPGM undergoes conformational exchange, which leads to the question whether you can confidently say that other states are not present at lower, but still significant, populations than that of the detected species. Would it be possible to determine the relative populations of phosphorylated cis and trans forms using ^{31}P NMR?

The reviewer is correct to point out that although the A: β G16BP complex is the only species that is observable in this experiment, other species (potentially other β G16BP complexes and/or E^{P} :G6P complexes) must have at least a threshold population to generate the observed ^1H , ^{15}N line-broadening. We have tried using ^{31}P NMR methods to observe signatures of these minor species but unfortunately found that we cannot observe the desired information. This is likely to be for a number of reasons. Firstly, the expected chemical shift differences for ^{31}P resonances from potential minor species are a problem. If the exchange is between A and B forms, the largest effect on ^{31}P chemical shifts from the *proximal* site is likely to be the engagement and dissociation of the positively charged amine group of K145. Our best estimate of the magnitude of this effect is the difference in ^{31}P chemical shifts of the 1-phosphate group of β G16BP in the $\beta\text{PGM}_{\text{D10N}}:\beta\text{G16BP}$ complex (Johnson et al., ACS Catal, 2018, 8, 8140–8153) in the presence and absence of the catalytic Mg^{2+} ion in the *proximal* site (where the positive charge coordinating the 1-phosphate group in the *proximal* site is changed). Here the resonance frequency difference is ca. 140 Hz. If the exchange is between an E: β G16BP complex and an E^{P} :G6P complex, our best estimate of the magnitude of this effect is the difference in ^{31}P chemical shifts of β G16BP (Johnson et al., ACS Catal, 2018, 8, 8140–8153) and an aspartyl phosphate in a peptide (e.g. in GGDA – Schlemmer et al., Magn Reson Chem, 1988, 26, 260–263), which corresponds to a frequency difference of ca. 540 Hz. Whatever the source, this exchange must be occurring at least as fast as k_{cat} or it would be rate limiting in a saturated complex. As k_{cat} is $\sim 400\text{ s}^{-1}$ under the conditions of the NMR experiment then exchange broadening of ^{31}P resonances associated with the major – minor transition is inevitable. Secondly, due to the sensitivity of ^{31}P , a very high

concentration of $\beta\text{PGM}_{\text{WT}}$ is required, which in turn requires an extremely high concentration of βG16BP (>100-fold excess of $\beta\text{PGM}_{\text{WT}}$ for the transient E: βG16BP complex to have sufficient lifetime for the observation of any ^{31}P signals). This presents a number of problems. Foremost amongst these is resolving ^{31}P peaks for minor species, such as A^{P} and B^{P} with or without bound G6P in the presence of large unbound βG16BP signals, and growing populations of G6P and P_i products. This is very challenging, particularly in the presence of Mg^{2+} ions, which cause line broadening of the free ^{31}P peaks. Hence, the combination of theoretical grounds for the experiment to fail to deliver the information sought, coupled with the technical difficulties associated with the experiments (including limited access to our laboratory due to Covid-19), means that we haven't chosen to go further down this path.

Related to this issue, you state in the abstract and elsewhere: "...b-glucose 1,6-bisphosphate, ..., phosphorylates both conformers into the more active species." It is not clear to me why you believe that both the A (cis) and B (trans) conformers become phosphorylated by βG16BP . If this is the case, then why does the kinetics not show a lag phase similar to that observed with F16BP? It seems to me that a more likely explanation would be that βG16BP can phosphorylate only the A conformer. It appears (but see above) that βG16BP binds with strong preference to the A conformer. Please clarify this issue.

If βG16BP simply phosphorylates *conformer A* and *conformer B*, generating A^{P} and B^{P} , respectively, a lag phase would indeed be observed in the kinetic profiles, while B^{P} (the inactive species) slowly converts to A^{P} (the active species). Alternatively, if βG16BP only phosphorylates *conformer A*, a lag phase would also be present in the kinetic profile arising from the slow conformational exchange of *conformer B* to *conformer A*, prior to phosphorylation. Our observations require that, in addition to the phosphorylation of *conformer A*, βG16BP phosphorylation of the enzyme population in *conformer B* is coupled with the *conformer B* to *conformer A* transition, resulting in the formation of A^{P} as the only phosphorylated species. Here, the βG16BP -mediated conversion rate must be faster than the phosphorylation rate for there to be a negligible population of B^{P} , resulting in the observed linear initial responses in the kinetic profiles, i.e. the absence of a lag phase. It is likely that the conversion of *conformer B* to *conformer A* occurs prior to or coincidentally with phosphorylation, since the presence of G6P and B^{P} does not reproduce the effect of βG16BP phosphorylation of *conformer B*. We have amended the text to clarify this argument, as follows:

ABSTRACT: *In vivo* phosphorylating agents, such as fructose 1,6-bisphosphate, generate phosphorylated forms of both conformers, leading to a lag phase in activity until the more active phosphorylated conformer dominates. In contrast, the reaction intermediate β -glucose 1,6-bisphosphate, which is at a very low concentration in the absence of β -glucose 1-phosphate, is able to couple the conformational switch and the phosphorylation step, resulting in the rapid generation of the more active phosphorylated conformer.

INTRODUCTION: Alternative phosphorylating agents such as F16BP and AcP generate phosphorylated forms of both conformers, resulting in a lag phase in βPGM activity until the more active phosphorylated conformer dominates. In contrast, the βG16BP reaction intermediate is able to couple the conformational switch and the phosphorylation step, resulting in the rapid generation of the more active phosphorylated species.

DISCUSSION: This result demonstrates that phosphorylation with βG16BP leads to the stabilisation of *conformer A*, resulting in production of A^{P} , regardless of the initial βPGM conformation, even for the

β PGM_{P146A} variant despite the considerable energetic cost of the *conformer B* to *conformer A* transition.

Continuing on this thread, would it be possible to determine the crystal structure of the D10N mutant in complex with F16BP? Perhaps this structure would reveal both cis and trans conformers in the complex. Perhaps modeling can reveal any differences in the binding modes of bG16BP and F16BP?

There appears to be a fundamental problem crystallising *conformer B*, unless *conformer A* is substantially destabilised. We suspect that this reflects the less ordered nature of *conformer B* with respect to *conformer A*, which is apparent in its NMR properties (e.g. RCI-S² values). For example, despite the measurable population of *conformer B* in solution conditions from which the substrate-free form can be crystallised, all structures (of more than 20 to-date in our lab alone, are *conformer A* (except for β PGM_{P146A})). Similarly, all GSA and TSA structures of β PGM are *conformer A*. Hence, we have every reason to believe that the crystal structure of the β PGM_{D10N}:F16BP complex will follow this pattern. In any case, our synchrotron access is only open for Covid-19 related work, currently.

Our modelling to-date has also not resolved the difference in properties between *conformer A* and *conformer B*, apart from gross effects like the dissociation of the sidechain of K145 from the active site when the K145-P146 peptide bond is in the *trans* form. Our current hypothesis is that β G16BP binding stimulates regional ordering within the protein, which couples to the *conformer B* to *conformer A* transition, whereas F16BP and AcP binding do not have this effect. Unfortunately, this kind of subtle effect, occurring on a millisecond timescale, is very difficult to model in a convincing manner.

2. In Fig. 5b, the kinetic profiles of the P146A mutant with bG16BP and AcP appear to be biphasic. Please comment on this.

In this figure, the second phase in the kinetic profiles is caused by two contributions. The first contributor relates to enzyme activity once the β G1P substrate has been depleted, where any remaining β G16BP is turned over slowly to G6P (monitored by the coupled assay), with the rate-limiting step being the hydrolysis of phospho-enzyme. Therefore, the kinetic profiles for the reactions with β G16BP increase steadily. Seeing as both β PGM_{WT} and β PGM_{P146A} have similar phospho-enzyme hydrolysis rates, and that the β PGM_{P146A} reaction was conducted at a 40-fold higher enzyme concentration compared to the β PGM_{WT} reaction, the remaining β G16BP is processed faster in the β PGM_{P146A} reaction, leading to a steeper gradient. The second contributor involves evaporation from the assay plate wells, which results in a concentration of the signal-generating NADH in the reaction sample. This effect is present in all of the kinetic profiles, yet because a longer time-frame is presented for the β PGM_{P146A} reactions, the slope appears more prominent. Text has been added to the figure caption to explain the cause of the second phase in the kinetic profiles, as follows:

Following β G1P substrate depletion, the kinetic profiles show an apparent increase in G6P concentration, which results from: (1) the concentration of the reaction ingredients through evaporation from the assay plate wells and (2) for the reactions recorded using β G16BP, the enzyme-dependent conversion of remaining β G16BP to G6P via β PGM^P, occurring at a rate proportional to the amount of enzyme.

3. Did I get it right that the crystal structure of 'apo' bPGM shows only the cis conformer? Why is this? Can the observation be explained by crystal contacts? Please comment on this.

Yes, all substrate-free β PGM_{WT} structures crystallise as *conformer A* (*cis* K145-P146 peptide bond) despite the clear population of *conformer B* in solution. Only substrate-free β PGM_{P146A} crystallises as

conformer B. We believe that there are two drivers for this. Firstly, crystallisation has only been successful from buffers containing very high anion concentrations (chloride plus acetate), which favour *conformer A* in solution. Secondly, the less ordered nature of *conformer B* with respect to *conformer A* (mentioned above) is a likely contributor to the difficulty to crystallise.

There are no crystal contacts in the area surrounding P146. The closest contacts appear at the N-terminal end of the loop containing P146, where packing may indeed have some local influence. However, this packing is unlikely to control the isomerisation state of the K145-P146 peptide bond, as conformational heterogeneity has been observed for residues A136–V141 across several structures where the *cis*-proline is invariant (e.g. PDB 2WHE and PDB 1ZOL).

Reviewer #2 (Remarks to the Author):

The manuscript by Wood et al. describes NMR-based, crystallographic and kinetic evaluation of the lag phase in the activity of the enzyme beta-phosphoglucosyltransferase (β PGM) from *L. lactis*. The interesting facet here is how an enzyme which is monomeric might be regulated without the need for an additional binding site or differential activator but by the substrate itself in the phosphorylation step of the mutase reaction. That is, the enzyme undergoes a conformational change (in this case *trans* to *cis*) which is better facilitated by some phosphorylating agents than others. Because there is no one method that shows that the structure observed is connected to function there are extensive experiments to tie together the results from structure and kinetics. The technical execution of the work is very good (although see some questions about kinetics and some problems with description of structures).

The main question in terms of potential impact of the findings comes from the main point of the manuscript which is the observation of what the authors dub allomorphy. Other work has described the adoption of discrete functional states that correlate with the enzyme's major conformational states and are redistributed in the presence of a regulatory effector (see Nikos S. Hatzakis, et al. (2012) Single Enzyme Studies Reveal the Existence of Discrete Functional States for Monomeric Enzymes and How They Are "Selected" upon Allosteric Regulation. *Journal of the American Chemical Society* 134 (22), 9296-9302 DOI: 10.1021/ja3011429). Another factor here is that the phosphorylating activator is the intermediate of the enzyme reaction, thus it is not completely unexpected to find that it has the most stabilizing or organizing effect on the enzyme and leads to population of the most active conformer.

Almost all fine control mechanisms of enzyme catalysis described to-date involve a variation of the population between discrete functional states. However, how the variation in population is controlled is fundamentally different between mechanisms (as, for example, illustrated by the red and green species in Figure 7).

In allostery (as exemplified by the reference quoted by the reviewer), reaction with (or binding of) an allosteric effector occurs outside of the active site and shifts the enzyme population from an inactive form to an active form, which stimulates the transformation of substrate to product at a rate determined by k_{cat} . Following exhaustion of the substrate, the enzyme population remains in the active form whilst the effector is present.

In contrast, in allomorphy, reactions with allomorphic activators (which are not the primary substrate) occur **inside** the active site. A full activator shifts the enzyme population fully from an inactive form to an active form, which stimulates the transformation of the primary substrate to product at a rate determined by k_{cat} . A partial activator is unable to shift the enzyme population from an inactive form to an active form, resulting in a slower overall catalytic rate (k'_{cat}). Following exhaustion of the allomorphic activator, the enzyme population returns slowly (~5 min for β PGM) to the original equilibrium position.

In allokairy, binding of the primary substrate inside the active site shifts the enzyme population from an inactive form to an active form, at an exchange rate (k_{ex}) that is similar to k_{cat} , resulting in time-dependent sigmoidal activity profiles (in allomorphy, the exchange rate (k_{ex}) between the two enzyme forms is much slower than k_{cat} , resulting in conventional hyperbolic activity profiles). Following exhaustion of the primary substrate, the enzyme population returns quickly (at a rate similar to k_{cat}) to the original equilibrium position.

We acknowledge that the discussion and caption to Figure 7 lacked clarity and therefore we have modified the text accordingly, as follows:

DISCUSSION: The multi-second substrate-dependent non-allosteric conformational exchange mediated through *cis-trans* proline isomerisation seen in β PGM represents a hitherto unidentified mechanism of post-translational enzyme regulation. This regulation mechanism relies on the existence of alternative pathways with different rate limiting steps, where the catalytic rate depends on the capacity of an activator, acting as a substrate, to bias the enzyme population towards the fastest pathway, by stabilising the most active conformation (Fig. 7). Similar to allostery and allokairy, this mechanism depends on the ability of the enzyme to adopt at least two conformations with distinct activities, but does not require the binding of an additional effector to the protein, nor an equivalence between the conformational exchange rate and the catalytic rate. We suggest the name “allomorphy” for this mechanism, from the greek *allos* meaning “other” and *morphe* meaning “form”, in keeping with the previously described allostery and allokairy mechanisms¹⁵.

CAPTION: In allostery, binding (or reaction) of an allosteric effector (purple rectangle) outside of the active site shifts the enzyme population from an inactive form (red circle and red profile) to an active form (green square and green profile), which stimulates the transformation of substrate (blue oval) to product (yellow triangle) at the catalytic rate (k_{cat} , green arrow). In allokairy, binding of substrate in the active site shifts the enzyme population from an inactive form to an active form, at an exchange rate (k_{ex}) that is similar to k_{cat} , resulting in time-dependent activity profiles (gradient of light green to dark green profiles). Following exhaustion of substrate, the enzyme population returns to the original equilibrium position. In allomorphy, reaction of the activating substrate, termed here allomorphic full activator (green hexagon), in the active site shifts the enzyme population from an inactive form to an active form, which stimulates the transformation of the native substrate (blue oval) to product (yellow triangle) at the maximal catalytic rate (k_{cat} , green arrow and green profile). However, reaction of alternatives substrates, termed here allomorphic partial activators (pink pentagon), in the active site are unable to shift the enzyme population from an inactive form to an active form, resulting in a slower overall catalytic rate (k'_{cat} , pink arrow and pink profile). The exchange rate (k_{ex}) between the two enzyme forms is much slower than k_{cat} . Following exhaustion of the allomorphic activator, the enzyme population returns to the original equilibrium position.

The authors may wish to consider the following comments (given in order of appearance; page numbers from pdf file for review).

P 7 line 155- “While β G16BP is able to phosphorylate both conformers into the more active species, alternative phosphorylation agents such as AcP and F16BP generate phosphorylated forms of both conformers, resulting in lower β PGM activity.” This statement - as a stand alone makes no sense- reword to distinguish what is meant by the two possibilities- they seem the same.

See the response above to a similar point raised by Reviewer #1 (second paragraph).

Page: 9- The authors state “The β PGMWT conformers are present in the spectra as a result of slow conformational exchange rather than as chemically distinct species, since the addition of 3 mM BeCl₂ and 10 mM NH₄F to the β PGMWT sample induced the population of a single β PGMWT:BeF₃ complex” Please clarify- is this conformer A or conformer B?

NMR analysis of ¹³C β chemical shifts (BMRB 17851) indicates that P146 in the β PGM_{WT}:BeF₃ complex adopts a *cis* K145-P146 peptide bond (*conformer A*). X-ray crystallography of this complex (PDB 2WFA) also shows a *cis* K145-P146 peptide bond (*conformer A*). Text has been inserted to clarify that the β PGM_{WT}:BeF₃ complex is an analogue of phosphorylated *conformer A*.

Page: 10- in the description of the crystal structure “Two monomeric chains are present in the asymmetric unit, both of which overlaid closely with a previously reported substrate-free β PGMWT structure (PDB 2WHE29: non-H atom RMSDs of 0.56 Å and 0.95 Å), thus both chains reflect the binding of citrate to only one of conformer A or conformer B.” This is very confusing- how can both chains reflect the binding of citrate to one conformer. Is citrate bound to one or both chains? This should be clarified. Also if citrate is bound to one or both- it means there is no change in structure as rmsd is small compared to unbound for both chains. How then is one chain designated as conformer A and the other as B? Are the authors saying that one conformer of the crystal structure is consistent with one set of NMR assignments (conformer A) and the other with conformer B ie – one is *cis* and the other *trans*? Please clarify.

We acknowledge that this section requires significant clarification and have reworked several sentences, as follows:

Two chains are present in the crystallographic asymmetric unit, one of which has citrate and acetate bound, whilst the other has tris and acetate bound. Citrate is coordinated in the active site by residues T16, H20, V47–R49 and A115–K117 and mimics substrate binding to some extent. Both monomers share a similar fold and overlay closely with a previously reported substrate-free β PGM_{WT} structure (PDB 2WHE²⁹: non-H atom RMSDs of 0.56 Å and 0.95 Å).

Page: 11 – The authors state after description of the NMR and the X-ray crystal structures “these experiments show that both conformer A and conformer B remain well-populated under near-physiological conditions.- the X-ray structures are performed under very different conditions than NMR (high salt and precipitant) so it is not apparent how both these experiments are under physiological conditions?

We agree that this summary sentence was not as accurate as it should have been. We have amended the text accordingly, as follows:

Although only one of the two conformers observed in solution is represented in the crystal, the NMR experiments show that both *conformer A* and *conformer B* remain well-populated under physiological conditions.

It is notable that the ¹³C chemical shift for P146 corroborates the presence of a *cis* K145-P146 peptide bond in solution for conformer A. However for conformer B, the absence of proline ¹³C resonances

for P146 precluded the identification of the trans isomerisation state for this proline. Thus the evidence for the trans conformer is indirect. However, the crystal structure together with the NMR results are consistent with the β PGMP146A structure being a close model of conformer B ie. the trans conformer

It is unfortunate that the $^{13}\text{C}\beta$ resonance of P146 in *conformer B* is not observable, thereby failing to provide direct evidence for a *trans* K145-P146 peptide bond. Considerable thought has been given as to methodologies that could provide chemical shifts for this region of *conformer B*. However, a combination of millisecond exchange in *conformer B* and multi-second exchange between *conformer A* and *conformer B* preclude a route to measure the relevant chemical shifts. We agree that the X-ray crystallography and NMR results are consistent with β PGM_{P146A} being a close model of *conformer B*.

Page: 13 The authors state "The D137-A147 loop also shows significant positional variation between different substrate-free β PGMWT structures (2WHE and 1ZOL... implying that conformer B is present in some β PGMWT crystal structures" but this is not precisely true - although the start of this segment (137-143) varies more between these two structures- after residue Ser144 they are nearly identical such that 144, 145 146 and 147 overlay very nicely. Although the NMR data is more convincing, the authors need to have better arguments to support their model that conformer B is present in some unliganded WT crystal structures

This sentence has been removed from the text, as we agree that the observation does not provide strong evidence for the population of *conformer B* in β PGM_{WT} crystal structures.

The authors state "In β PGMWT, the ϵ -amino group of K145 is coordinated by the carbonyl oxygen atom of A113, the carboxylate side chain of E169, and a water molecule that is replaced in the transition state analogue (TSA) complex (PDB 2WF529) by a fluoride ion that mimics an oxygen atom of the transferring phosphoryl group." But looking at the structure - it is also making a bond to the catalytically essential Asp 8. This bond is consistent between the unliganded and liganded TSA forms and should be noted here.

In accordance with standard C-N-H bond angles, the three hydrogen atoms of the ϵ -amine group of K145 can be positioned to form regular hydrogen bonds to the oxygen atom of A113, the carboxylate sidechain of E169, and a water molecule (or a fluoride atom of MgF_3^- or an oxygen atom of the transferring phosphoryl group). The reviewer is correct to point out that, additionally, the sidechain nitrogen atom of K145 is close to a carboxylate oxygen atom of D8 ($\sim 3.1 \text{ \AA}$), and therefore there will be an electrostatic relationship between these groups, even though the relative position of the heavy atoms does not fit with a regular hydrogen bond. This proximity has been added to the text, as follows:

An electrostatic relationship also exists between the ϵ -amino group and the carboxylate group of D8.

Page: 15- The authors calculate the kinetics for the P146A variant and find "Initial rate measurements were made at several β G1P and β G16BP concentrations (10-500 μM and 0.4-100 μM respectively) and were globally fit to the equation used for β PGMWT above. The fitting yielded values for k_{cat} of $19.2 \pm 0.2 \text{ s}^{-1}$, K_{m} (β G1P) of $157 \pm 3 \mu\text{M}$, K_{m} (β G16BP) of $175 \pm 3 \mu\text{M}$ for β PGMP146A" A possible problem here is that in this series of experiments 10 μM β G16BP was used as phosphorylating agent but that is ten fold below the calculated K_{m} of 157 so little of the enzyme may be phosphorylated simply due to binding of the agent.

The initial rate measurements used for calculating the kinetic parameters of β PGM_{P146A} were acquired at a range of β G16BP concentrations (2, 5, 10, 35, 50, 100 μM) and not just at 10 μM β G16BP. If the reviewer is referring to the conditions used to generate the kinetic profile presented in Figure 5, then

it is true that 10 μM βG16BP lies below K_m for βG16BP . However, these conditions were chosen to ensure that all of the curves in the figure plateaued at similar product (G6P) concentrations, whilst also illustrating the absence of a lag phase in the βG16BP kinetic profile. Similar kinetic profiles showing a linear initial response are observed when using higher βG16BP concentrations, but with a corresponding increase in the amplitude of the curve. We acknowledge that the details for the coupled assay experiments were not provided with sufficient clarity and therefore we have amended the text in the Results and Methods sections together with the caption to Supplementary Figure 9, as follows:

RESULTS: In the coupled assay experiments with $\beta\text{PGM}_{\text{WT}}$ (Fig. 5a), when either F16BP or AcP was used as the phosphorylating agent, their progression curves display significant lag phases. The lag is considerably more pronounced in the F16BP experiment, and consequently the maximum rate of βG1P to G6P conversion is not achieved before the substrate is exhausted. When βG16BP was used as the phosphorylating agent the kinetic profile shows a linear, fast initial rate. Consequently, initial rate measurements were made at several βG1P and βG16BP concentrations (10–700 μM and 0.4–100 μM , respectively) and were globally fitted to an equation derived for a ping-pong mechanism with βG1P inhibition²⁸.

RESULTS: As for $\beta\text{PGM}_{\text{WT}}$, the kinetic profiles for $\beta\text{PGM}_{\text{P146A}}$ display significant lag phases with F16BP and AcP, while the progression curves with βG16BP show a linear, fast initial rate (Fig. 5b). Consequently, initial rate measurements were made at several βG1P and βG16BP concentrations (5–500 μM and 2–100 μM respectively) and were globally fit to the equation used for $\beta\text{PGM}_{\text{WT}}$ above.

METHODS: For the determination of k_{cat} and K_m values for $\beta\text{PGM}_{\text{WT}}$, the reaction was initiated by dilution of the enzyme prepared in standard kinetic buffer to a final concentration of 1 nM $\beta\text{PGM}_{\text{WT}}$ in solutions of 1 mM NAD^+ and 5 units mL^{-1} G6PDH and variable concentrations of βG1P (10, 20, 30, 50, 70, 100, 150, 200, 300, 500, 700 μM) and βG16BP (0.4, 1, 2, 5, 10 μM). For the determination of k_{cat} and K_m values for $\beta\text{PGM}_{\text{P146A}}$, the reaction was initiated by dilution of the enzyme prepared in standard kinetic buffer to a final concentration of 100 nM $\beta\text{PGM}_{\text{P146A}}$ in solutions of 1 mM NAD^+ and 5 units mL^{-1} G6PDH and variable concentrations of βG1P (5, 10, 15, 20, 30, 50, 70, 100, 200, 300, 500 μM) and βG16BP (2, 5, 10, 35, 50, 100 μM).

CAPTION: (a) $\beta\text{PGM}_{\text{WT}}$ initial rate measurements at a range of βG1P concentrations (10, 20, 30, 50, 70, 100, 150, 200, 300, 500, 700 μM) and βG16BP concentrations (0.4, 1, 2, 5, 10 μM , grey gradient increasing with concentration). (b) $\beta\text{PGM}_{\text{P146A}}$ initial rate measurements at a range of βG1P concentrations (5, 10, 15, 20, 30, 50, 70, 100, 200, 300, 500 μM) and βG16BP concentrations (2, 5, 10, 35, 50, 100 μM , green gradient increasing with concentration).

Page: 16- The authors state "Additionally, the increase in K_m for βG16BP in $\beta\text{PGM}_{\text{P146A}}$ reflects the energetic cost of the trans to cis isomerisation of the K145-A146 peptide bond⁴⁶." This may be true if K_m represents a dissociation constant for βG16BP , but as K_m also includes rate constant(s), then the difference in K_m cannot be equated with this energy. If the authors have an argument to show that K_m equals K_d then it should be given here.

Assuming that the βG16BP association rate is diffusion controlled ($k_{\text{on}} \sim 10^9 \text{ M}^{-1} \text{ s}^{-1}$) and $K_m = (k_{\text{off}} + k_{\text{cat}})/k_{\text{on}}$, then k_{off} can be estimated as 22-fold and $\sim 10^5$ -fold larger than k_{cat} for $\beta\text{PGM}_{\text{WT}}$ and $\beta\text{PGM}_{\text{P146A}}$, respectively. With $k_{\text{off}} \gg k_{\text{cat}}$, then $K_d \sim K_m$ in this approximation. The estimated K_d values, 8.1 μM and 175 μM , are close to within the experimental error of their respective K_m values, $8.5 \pm 0.3 \mu\text{M}$ and $175 \pm 3 \mu\text{M}$, for $\beta\text{PGM}_{\text{WT}}$ and $\beta\text{PGM}_{\text{P146A}}$. This argument gives us confidence in using the experimental K_m

values as a proxy for β G16BP binding constants. We have expanded the explanation in the text, as follows:

Additionally, assuming that β G16BP binding is diffusion controlled, the increase in K_m for β G16BP in β PGM_{P146A} reflects the energetic cost of the *trans* to *cis* isomerisation of the K145-A146 peptide bond.

Page: 17- The rate constant for dephosphorylation (through hydrolysis) was determined to be $0.02 \pm 0.002 \text{ s}^{-1}$ from the rate of reduction of the free F16BP concentration in ^1H NMR experiments. - is this the correct statement?- it seems that the rate of disappearance of the F16BP would equate with the rate of enzyme phosphorylation not of dephosphorylation. When the phosphoryl group is removed from F16BP the enzyme is phosphorylated and dephosphorylation ensues from the hydrolysis of the enzyme

For β PGM_{WT}, the dominant steady state species at 50 mM F16BP is phosphorylated *conformer A* (A^P). This observation is consistent with the rate of dephosphorylation of A^P (through phospho-enzyme hydrolysis) being the rate-limiting step of F16BP disappearance monitored by ^1H NMR experiments. Phosphoryl transfer from F16BP to D8 cannot occur in A^P until the aspartyl-phosphate moiety has been hydrolysed. Given the signal-to-noise ratio of the NMR spectra, the rate constant of phosphorylation from F16BP to D8 generating A^P is at least an order of magnitude larger than the rate constant of dephosphorylation of A^P . Hence, the phosphorylation rate cannot be accurately determined using these experiments. In contrast, for β PGM_{P146A}, the primary steady state species are substrate-free β PGM_{P146A} (*conformer B*) and phospho-enzyme (B^P). The rate constant of *conformer B* phosphorylation by F16BP and the rate constant of dephosphorylation of B^P are very similar, and the experimental error precludes any further deconvolution. We have clarified this limitation in the text, as follows:

The presence of *conformer B* shows that the rate of β PGM_{P146A} phosphorylation is very similar to the dephosphorylation rate for B^P (through hydrolysis), and only an apparent rate constant can be measured. The apparent rate constant for dephosphorylation was determined to be $0.02 \pm 0.002 \text{ s}^{-1}$ from the rate of reduction of the free F16BP concentration in ^1H NMR experiments.

Page: 22- The multi-second conformational exchange due to *cis-trans* proline isomerisation seen in the substrate-free form of β PGM represents a hitherto unidentified mechanism of post-translational enzyme regulation. As described in this reviewers opening statements, I believe this mechanism has been previously observed for monomeric enzymes.

See the response above to the general comment raised by Reviewer #2.

ref 10 and 32 are the same

This comment relates to a fully referenced abstract contained in the PDF file, which was a requirement for submission to *Nature*. The references in the abstract have now been removed following the *Nature Communications* guidelines for authors.

Reviewer #3 (Remarks to the Author):

This manuscript describes an extensive experimental investigation into β -phosphoglucomutase, which results in the proposal of a novel regulatory mechanism that the authors call "allomorphy". NMR and x-ray crystallographic methods have been used to characterize two different conformers of β -PGM, resulting from isomerization of a peptide bond, and the authors offer convincing arguments that they serve a biological purpose. The enzyme has a short-lived phosphorylated form that is part of the

catalytic cycle. A high activity enzyme form is stabilized when the phosphorylated enzyme is formed from the natural substrate; alternative phosphorylating agents result in the preponderant formation of a lower activity conformer.

Allomorphy is a novel concept, and the experimental data as interpreted by the authors provide solid support for it. It is a rather subtle concept, however, and moving Figure 7 to the introduction would better prepare the reader for the arguments that follow.

As requested by the Editor, Figure 7 will remain in the current position in the Discussion.

Figure 5b shows a significant slow phase following the initial rapid phase when the P146A enzyme is phosphorylated by acetyl-P or β -G16BP. Do the authors know what causes this?

See the response above to point 2 raised by Reviewer #1.

Overall, this is a well-written manuscript, and the following are minor points.
-line 40. "...phosphorylates both conformers into the more active species." This is odd phrasing; a compound is phosphorylated, not phosphorylated *into* something.

See the response above to a similar point raised by Reviewer #1 (second paragraph).

-lines 110-114. Phosphorylases catalyze phosphorolysis reactions, not hydrolysis by a Pi-dependent enzyme.

The text and figure caption have been amended accordingly.

-Figure 6. What is the meaning of the double-headed arrows that connect BP and AP?

The following text has been added to the figure caption:

The double-headed arrows connecting A^P and B^P indicate that these species interconvert with a multi-second exchange rate, similar to that described for the interconversion of *conformer A* and *conformer B*.

Peter Tipton

Reviewer #4 (Remarks to the Author):

Wood et al describe a very thorough analysis of the unusual regulatory mechanism in an enzyme, β -phosphoglucosylmutase, of the basic metabolism. This mutase functions as a phosphoenzyme. The phosphate group tends to easily hydrolyze. Therefore, the enzyme needs a continuous supply of phosphate donors that, physiologically, can be β -glucose 1,6-bisphosphate (a dissociable reaction intermediate) or fructose-1,6-bisphosphate. The authors demonstrate that these two compounds are not at all equivalent. β -Glucose 1,6-bisphosphate efficiently phosphorylate the enzyme and promotes its activate conformation. By contrast, fructose-1,6-bisphosphate less efficiently modifies the enzyme and hardly promotes its conversion to the active conformation. Indeed, the NMR, crystallographic, and mutagenesis studies described in this manuscript demonstrate the existence of inactive and active states and interconversion between the two forms can limit the reaction and cause a lag phase. In other words,

β -glucose 1,6-bisphosphate primes the enzyme conformation for catalysis whereas fructose-1,6-bisphosphate does not so. The conformational change involves the isomerization of a peptide bond. The idea that the substrate (or its analogue) can alter the equilibrium between two protein conformations is not new in enzymology; certain enzymes seem to keep a memory of substrate binding so that once they start to turn substrates, they increase their reactivity. The value of this manuscript is in the depth of the analysis and the implications of these findings for regulation of metabolism. Therefore, I think that the manuscript is a valuable contribution with some new insight into a “classical” problem of enzymology.

We thank the reviewer for their comments regarding the manuscript having ‘some new insights into a “classical” problem of enzymology’. The crux of the issue here is that as long ago as the 1960s it was clear that some enzymes showed hysteretic behaviour when exposed to substrate, but were not classical allosteric systems (Freiden, *Ann Rev Biochem*, 1979, 48, 471–489). However, although the phenomenon was widely recognized, the underlying mechanism(s) were primarily theoretical models. Indeed, it was generally believed that observed hysteretic effects were entirely kinetic (Hilser et al., *Proc Natl Acad Sci USA*, 2015, 112, 11430–11431), and the recent demonstration of allokairy in glucokinase (Whittington et al., *Proc Natl Acad Sci USA*, 2015, 112, 11553–11558) was the first fully detailed experimental verification of such a model. However, allomorphy, as demonstrated here, is a completely different way of delivering hysteretic behaviour, one which does not fit the criteria of being entirely kinetic. Without repeating all of the details in our first response to Reviewer #2, allokairy is based on the similarity of rate for a conformational switch (between a less active and a more active form) and k_{cat} , and thus is a temporal mechanism. Allomorphy is based on different activating substrates leading to different levels of switching between less active and more active forms. There is no requirement for the switching rate to be fast. Hence, allomorphy is fundamentally a different mechanism to allokairy (and, of course, allostery). We expanded the discussion to reflect this, as follows:

Allomorphy may modulate the activity of other monomeric enzymes with hysteretic behaviour, i.e. those that exhibit a burst or lag phase in their kinetic profile⁴⁹. Several theoretical models have been put forward to rationalise hysteretic behaviour, such as the mnemonic⁵⁰ and the ligand-induced slow transition⁵¹ models, but detailed structural-based molecular mechanisms have proved elusive. To our knowledge, only one such mechanism, allokairy in human glucokinase, has been described in detail^{15,16}. Allomorphy is a different fine control regulatory mechanism and is potentially widespread, at least across phosphomutases; for example, both rabbit muscle and *L. lactis* α -phosphoglucosmutases appear to be hysteretic enzymes^{52,53}, but belong to very different protein superfamilies. Like β PGM, these enzymes require a phosphorylating agent to initiate the catalytic cycle and, for the latter, the use of the reaction intermediate results in linear kinetics, whereas alternative phosphorylating agents produce a lag phase in their kinetic profiles. Similarly, α -phosphomannomutase from *Galdieria sulphuraria*, which also requires the addition of a phosphorylating agent to initiate the catalytic cycle, exhibits linear kinetics when α -mannose 1-phosphate and α -mannose 1,6-bisphosphate (or α -glucose 1-phosphate and α G16BP) are included in the reaction, but has a lag phase when there is a mismatch between substrate and phosphorylating agent, or when F16BP is used as the phosphorylating agent⁵⁴. All of these observations are consistent with the presence of allomorphic control.

I have a few points:

-Line 212: this is a critical point. The authors use milk to mimic the physiological environment. Are we sure that this is indeed the case? Why should milk mimic the cytosol? This point is particularly important in light of the data reported in line 206: 200 mM NaCl trigger the active conformation. This salt concentration does not seem too far from a “physiological” ionic strength. This point needs to be

discussed to strengthen the idea that the observed kinetic/regulatory effects can indeed be relevant for metabolism regulation in the cell.

We acknowledge that the rationale behind the use of bovine milk to mimic the intracellular environment of *L. lactis* was not outlined with enough clarity. *L. lactis*, along with many other species of Gram-positive bacteria, control their intracellular environments using mechanosensitive channels and osmoregulation transport systems, together with osmoprotectant compounds to actively regulate osmolyte concentrations so that turgor is maintained within a specific range. *L. lactis* cytoplasm contains elevated concentrations of K^+ (~500 mM) as part of the control of intracellular pH levels. In β PGM_{WT}, the population distribution between *conformer A* and *conformer B* is not significantly affected by the presence of 200 mM K^+ HEPES buffer. The concentration of Na^+ in the cytoplasm has been determined to be ~50 mM and concentrations as high as 200 mM Na^+ are not regarded as physiological from studies optimising the components of an *in vivo*-like assay medium. Moreover, changes in the *conformer A* and *conformer B* equilibrium that the reviewer highlights were also observed at 100 mM $MgCl_2$, indicating that the primary effector is Cl^- anions, which are present in *L. lactis* cytoplasm with a concentration of ~50 mM, where *conformer B* remains significantly populated.

However, the intracellular milieu is a complex mix of metabolites that could influence this equilibrium. This property of the environment was mimicked through the use of bovine skimmed milk, a medium in which *L. lactis* thrives within the dairy industry. It is expected that the organic components in milk will also be present in cytoplasm. Since the inorganic ionic composition (~5 mM Mg^{2+} , ~24 mM Na^+ , ~38 mM K^+ , ~28 mM Cl^-) of milk is similar to cytoplasm (except for K^+ , which has no effect on the equilibrium between *conformer A* and *conformer B*), any effects will be due to the influence of metabolites. The text has been amended to provide additional clarity, as follows:

An investigation of factors that affect the population distribution of *conformer A* and *conformer B* was performed using $^1H^{15}N$ -TROSY spectra of β PGM_{WT} recorded under different conditions of temperature, pH, hydrostatic pressure, $MgCl_2$ (0–100 mM), $NaCl$ (0–200 mM), K^+ HEPES buffer (0–200 mM) and β PGM_{WT} concentration (0.1–1.2 mM). All of these perturbations had little or no effect, apart from the addition of either $MgCl_2$ (100 mM) or $NaCl$ (200 mM) to standard NMR buffer, which shifted the population of β PGM_{WT} primarily to *conformer A* (Supplementary Fig. 6a–d). Buffer exchange into deionised water resulted in *conformer B* being the dominant population. However, both *conformer A* and *conformer B* remained populated when Mg^{2+} was removed from the NMR buffer solution, showing that the multi-second conformational exchange is not simply a result of incomplete saturation of the catalytic Mg^{2+} binding site. These observations indicate that chloride anions perturb the population distribution.

The inorganic ionic composition of *L. lactis* cytoplasm (~2 mM Mg^{2+} , ~50 mM Na^+ , ~400 mM K^+ , ~50 mM Cl^-)³⁷ overlaps with the concentration ranges tested, where the population distribution between *conformer A* and *conformer B* remained unaffected. Therefore, it is expected that both *conformer A* and *conformer B* are populated in cytoplasm. However, the intracellular milieu is a complex mix of metabolites that could influence this equilibrium. This environment was mimicked through the use of bovine skimmed milk, a medium in which *L. lactis* thrives within the dairy industry. It is expected that the organic components in milk will also be present in cytoplasm. Moreover, the inorganic ionic composition (~5 mM Mg^{2+} , ~24 mM Na^+ , ~38 mM K^+ , ~28 mM Cl^-)³⁸ is similar to cytoplasm (except for K^+ , which has no effect on the equilibrium between *conformer A* and *conformer B*), so any effects will be due to the influence of metabolites. β PGM_{WT} was diluted 5-fold into fresh skimmed milk, which had been filtered to remove species with a molecular weight larger than 10 kDa.

-Another critical point concerns the electron density of Figure 4C, which is not convincing. As it is represented, one would guess that the peptide is in multiple conformations rather than in a single *trans* conformation.

For the region encompassing the E140–P148 loop in $\beta\text{PGM}_{\text{P146A}}$, there are undoubtedly multiple conformations present in fast exchange in solution, as evidenced by the lower predicted RCI-S² values derived from NMR chemical shifts. These multiple conformations do not involve a switching between the isomerisation states of the K145–A146 peptide bond, as evidenced by a single species being observed in NMR spectra. This conformational flexibility is reflected in the weak electron density and the elevated temperature factors for these residues in the crystal structure of $\beta\text{PGM}_{\text{P146A}}$, particularly for the residues located N-terminally of K145. Taken in context with the residues positioned C-terminally of A146, which have much stronger electron density, the single chain, as modelled including a *trans* K145–A146 peptide bond, is justified and gives the best fit to the electron density. We have included two expanded omit maps (generated by refinement in the absence of residues S144–P148) of substrate-free $\beta\text{PGM}_{\text{P146A}}$ and the $\beta\text{PGM}_{\text{P146A}}:\text{MgF}_3:\text{G6P}$ TSA complex to support the interpretation of the electron density in Figure 4. These panels provide a clearer comparison between a *trans* K145–A146 peptide bond and a *cis* K145–A146 peptide bond in the substrate-free $\beta\text{PGM}_{\text{P146A}}$ and $\beta\text{PGM}_{\text{P146A}}:\text{MgF}_3:\text{G6P}$ TSA complex, respectively, within the S144–P148 segment. We have expanded the text and changed the figure caption accordingly, as follows:

RESULTS: The D137–A147 loop exhibits elevated temperature factors, consistent with the lower predicted RCI-S² values derived from NMR chemical shifts (Supplementary Fig. 5c). The electron density is best fit by the *trans* conformation of the K145–A146 peptide bond (ω dihedral angle = -177°) (Fig. 4a, c). In comparison to $\beta\text{PGM}_{\text{WT}}$, the D137–A147 loop adopts a different conformation, although both a 3_{10} -helix (D137–V141) and a β -turn hydrogen bond (A147_{HN}–S144_{CO}) are retained.

CAPTION: **c, d,** Omit map generated by refinement in the absence of residues S144–P148 in $\beta\text{PGM}_{\text{P146A}}$. **(c)** The S144–P148 segment, containing a *trans* K145–A146 peptide bond, with positive difference density ($F_o - F_c$; green mesh contoured at $+2.5\sigma$) in substrate-free $\beta\text{PGM}_{\text{P146A}}$. **(d)** The S144–P148 segment, containing a *cis* K145–A146 peptide bond, with positive difference density ($F_o - F_c$; green mesh contoured at $+2.5\sigma$) in the $\beta\text{PGM}_{\text{P146A}}:\text{MgF}_3:\text{G6P}$ TSA complex.

-I hate to suggest additional experiments. But in this case, a simple experiment that could be done (e.g. with a nanotemper instrument) would be to measure the thermal stability of the WT and mutant proteins in their different states (native, phosphorylated, activated, transition-state like as used for crystallization etc). This would give an idea about the effect of the *cis-trans* isomerization on protein stability in the WT and the Pro mutant.

This is an interesting suggestion but unfortunately we do not have access to such equipment currently. Also, it is likely that interpretation will be difficult for many of our systems as at least one of the components in a comparison exists as either mixed populations (e.g. substrate-free $\beta\text{PGM}_{\text{WT}}$ is a mixed population of *conformer A* and *conformer B*, and phosphorylated $\beta\text{PGM}_{\text{P146A}}$ is a mixed population of *conformer B* and *B'*) or involve a protein-ligand complex where one of the moieties cannot be defined when free in solution (e.g. MgF_3^- in the $\beta\text{PGM}_{\text{WT}}:\text{MgF}_3:\text{G6P}$ and $\beta\text{PGM}_{\text{P146A}}:\text{MgF}_3:\text{G6P}$ TSA complexes). Also, unfortunately, $\beta\text{PGM}_{\text{WT}}$ and $\beta\text{PGM}_{\text{P146A}}$ are prone to aggregation at elevated temperatures, which further complicates any meaningful comparison. In any case, linking melting temperatures (T_m values) to thermodynamic stabilities at normal operating temperatures requires making assumptions relating to heat capacity changes on unfolding, which would require very extensive experimental investigations to show whether they were valid. Instead, we prefer to simply comment on the

differences in free energy values between some species from observable populations in NMR experiments. Substrate-free $\beta\text{PGM}_{\text{WT}}$ is populated 70% *conformer A* and 30% *conformer B* (in standard NMR buffer) and is populated 60% *conformer A* and 40% *conformer B* under near-physiological conditions. This observation indicates that the free energy difference between *conformer A* and *conformer B* is 1–2 kJ mol⁻¹. Phosphorylation of $\beta\text{PGM}_{\text{WT}}$ with either AcP or F16BP eventually results in the almost exclusive population of A^{P} , indicating that this species has a significantly lower free energy value than B^{P} , *conformer A* or *conformer B* under the conditions of the experiment. Similarly, addition of BeCl₂ and NaF to $\beta\text{PGM}_{\text{WT}}$ results in the sole population of the $\beta\text{PGM}_{\text{WT}}:\text{BeF}_3$ complex, which is an analogue of A^{P} . Furthermore, addition of G6P and NaF to substrate-free $\beta\text{PGM}_{\text{WT}}$ (a mix of *conformer A* and *conformer B*) and $\beta\text{PGM}_{\text{P146A}}$ (overwhelmingly *conformer B*) generates $\beta\text{PGM}_{\text{WT}}:\text{MgF}_3:\text{G6P}$ and $\beta\text{PGM}_{\text{P146A}}:\text{MgF}_3:\text{G6P}$ TSA complexes in solution that both possess a *cis* K145-X146 peptide bond (i.e. *conformer A* like), which indicates that the TSA complexes have a significantly lower free energy than *conformer B* like forms in these complexes.

REVIEWERS' COMMENTS

Reviewer #1 (Remarks to the Author):

The authors have addressed all of the issues I raised in my previous review. The explanations given in the rebuttal letter are clear and satisfactory. The revised manuscript includes additions, with an expanded discussion, that resolve previously unclear statements.

Reviewer #3 (Remarks to the Author):

In the revised manuscript the authors have addressed all of the concerns raised previously. The revised manuscript is suitable for publication.

Reviewer #4 (Remarks to the Author):

I am impressed by the thorough revision of the manuscript in response to the Reviewers' comments. The newly added paragraphs at the end of the manuscript provide an excellent take-home message. The electron densities of Figures 4C-D are now more convincing. The responses to the comments about the NMR experiments are generally satisfactory. In my view, this is an excellent manuscript.

REVIEWERS' COMMENTS

Reviewer #1 (Remarks to the Author):

The authors have addressed all of the issues I raised in my previous review. The explanations given in the rebuttal letter are clear and satisfactory. The revised manuscript includes additions, with an expanded discussion, that resolve previously unclear statements.

Reviewer #3 (Remarks to the Author):

In the revised manuscript the authors have addressed all of the concerns raised previously. The revised manuscript is suitable for publication.

Reviewer #4 (Remarks to the Author):

I am impressed by the thorough revision of the manuscript in response to the Reviewers' comments. The newly added paragraphs at the end of the manuscript provide an excellent take-home message. The electron densities of Figures 4C-D are now more convincing. The responses to the comments about the NMR experiments are generally satisfactory. In my view, this is an excellent manuscript.

We would like to thank all the reviewers for their positive comments regarding the revision of the manuscript.